# Evolution of a guarded decoy protease and its receptor in solanaceous plants

Jiorgos Kourelis [1], Shivani Malik[1], Oliver Mattinson [1], Sonja Krauter[1], Parvinderdeep S. Kahlon[1], Judith K. Paulus[1] & Renier A. L. van der Hoorn [1]✉

Rcr3 is a secreted protease of tomato that is targeted by fungal effector Avr2, a secreted protease inhibitor of the fungal pathogen *Cladosporium fulvum*. The Avr2-Rcr3 complex is recognized by receptor-like protein Cf-2, triggering hypersensitive cell death (HR) and disease resistance. Avr2 also targets Rcr3 paralog Pip1, which is not required for Avr2 recognition but contributes to basal resistance. Thus, Rcr3 acts as a guarded decoy in this interaction, trapping the fungus into a recognition event. Here we show that Rcr3 evolved > 50 million years ago (Mya), whereas Cf-2 evolved <6Mya by co-opting the pre-existing Rcr3 in the *Solanum* genus. Ancient Rcr3 homologs present in tomato, potato, eggplants, pepper, petunia and tobacco can be inhibited by Avr2 with the exception of tobacco Rcr3. Four variant residues in Rcr3 promote Avr2 inhibition, but the Rcr3 that co-evolved with Cf-2 lacks three of these residues, indicating that the Rcr3 co-receptor is suboptimal for Avr2 binding. Pepper Rcr3 triggers HR with Cf-2 and Avr2 when engineered for enhanced inhibition by Avr2. *Nicotiana benthamiana (Nb)* is a natural null mutant carrying *Rcr3* and *Pip1* alleles with deleterious frame-shift mutations. Resurrected *Nb*Rcr3 and *Nb*Pip1 alleles were active proteases and further *Nb*Rcr3 engineering facilitated Avr2 inhibition, uncoupled from HR signalling. The evolution of a receptor co-opting a conserved pathogen target contrasts with other indirect pathogen recognition mechanisms.

[1] Plant Chemetics Laboratory, Department of Plant Sciences, University of Oxford, South Parks Road, OX1 3RB Oxford, UK. ✉email: renier.vanderhoorn@plants.ox.ac.uk

Plants recognize pathogens using both cell-surface and intracellular immune receptors. Many of these receptors are encoded by polymorphic resistance (R) genes. The products of these R genes function either by directly recognizing pathogen components, or by indirectly recognizing the interaction of pathogen components with host components or modification of these host components by pathogen effectors[1]. This second, indirect mechanism is frequently explained by the 'guard' or 'decoy' model, in which a host component is either monitored—'guarded'—by a R gene product or in which a host component mimics an operative target of the pathogen, trapping it into a recognition event[2].

Receptors encoded by R genes typically belong to two classes of receptors: (i) cell-surface receptor-like proteins (RLPs) and receptor-like kinases (RLKs), and (ii) intracellular Nod-like receptors (NLRs). NLRs can contain additional integrated domains (IDs), some of which were shown to interact with pathogen elicitors, thereby triggering immune responses[3]. How a guard/decoy mechanism evolves in genetically unlinked receptor/pathogen target, however, is not well understood. One distinct, well-studied example of a genetically unlinked guard/decoy mechanism is the recognition of the tomato leaf mould causing pathogen Cladosporium fulvum (syn. Passalora fulva) by the product of the tomato R gene Cf-2.

Cf-2 was introgressed into cultivated tomato (Solanum lycopersicum) from the wild currant tomato (Solanum pimpinellifolium) and encodes for a RLP with extracellular leucine-rich-repeats (LRRs)[4]. Cf-2 recognizes C. fulvum strains producing a secreted small cysteine-rich effector called Avr2[5]. This recognition also requires Rcr3, which encodes for a papain-like cysteine protease (PLCP)[6]. Avr2 acts as a protease inhibitor, and it is the interaction of Avr2 with Rcr3, which is recognized by Cf-2 resulting in a hypersensitive response (HR) and immunity to C. fulvum[7].

Besides Rcr3, Avr2 also inhibits immune protease Pip1, a paralog of Rcr3[8]. Both Rcr3 and Pip1 are regulated as pathogenesis-related (PR) genes[8], but Pip1 protein is more abundant in the apoplast than Rcr3[9,10]. While knockout of Rcr3 does not result in enhanced susceptibility to C. fulvum in tomato lines lacking Cf-2, knockdown of Pip1 strongly increases susceptibility to C. fulvum[10]. This indicates that during infection with C. fulvum, Pip1 is the actual operative target of Avr2, while Rcr3 acts as a decoy to trap the pathogen into a recognition event[2]. However, plants lacking Rcr3 are more susceptible to the oomycete late blight pathogen Phytophthora infestans, even in the absence of Cf-2[8,10], indicating that Rcr3 can also provide immunity in the absence of Cf-2.

Besides Avr2, Rcr3 is also inhibited by GrVAP1, a secreted protein of the potato cyst nematode Globodera rostochiensis, which is unrelated to Avr2 but still triggers HR in Cf-2 tomato plants[11]. Rcr3 is also inhibited by cystatin-like EPIC1 and EPIC2B of the oomycete late blight pathogen Phytophthora infestans and the chagasin-like Cip1 of the bacterial leaf spot pathogen Pseudomonas syringae[8,12,13]. These EPIC and Cip1 inhibitors, however, do not trigger Cf-2-dependent HR and are considered stealthy inhibitors that manipulate the host without being detected[8,13].

Given its multifaceted role in immunity, we hypothesise that Rcr3 will be under different selection pressures acting on three different regions in the protein. First, substrate recognition in the substrate-binding groove is important for Rcr3 function as an immune protease in Cf-2 independent immunity. Second, there will be pressure on Rcr3 caused by pathogen-derived inhibitors in two opposite directions. In the absence of Cf-2, one would expect selection pressure to avoid inhibition to maintain Rcr3 function in basal immunity. By contrast, in the presence of Cf-2, one would expect selection to increase interactions with inhibitors to maximise perception sensitivity and specificity. Furthermore, in the presence of Cf-2, there will be very strong selection on Rcr3 to prevent auto-necrosis induced, e.g., by unintended interactions with endogenous inhibitors[6]. Interestingly, however, because the structures of the different pathogen-derived inhibitors are unrelated, one would expect multiple variant residues affecting different inhibitor interactions in complex ways. Third and finally, interactions with Cf-2 will put selection on Rcr3 but only in plant species that contain Cf-2.

Here we investigated the evolution of the Cf-2/Rcr3 perception system in solanaceous plant species to determine which gene evolved first, using a newly established transient HR assay to study Avr2/Rcr3/Cf-2 by agroinfiltration in N. benthamiana. We also studied the role of variant residues in Rcr3 in Avr2 inhibition and HR signalling. This work provides new insights into the evolution of a guarded decoy and its receptor in Solanaceae.

## Results

**Co-expression of Cf-2, Rcr3, and Avr2 triggers HR**. To accelerate our studies on Rcr3 evolution, we developed a transient HR assay for Cf-2-dependent recognition of Avr2 in N. benthamiana by co-infiltration of Agrobacterium strains containing binary vectors for the expression of the S. pimpinellifolium Rcr3 allele (SpRcr3), Avr2, and Cf-2. We used a synthetic Cf-2 construct in which the repetitive repeats present in the natural DNA sequence were removed to prevent recombination in E. coli and A. tumefaciens during cloning and transient expression. Co-infiltration of these strains in the leaves of N. benthamiana triggers HR, which starts at 2–3-days post-infiltration (dpi) and is fully developed within 5 dpi (Fig. 1a). No HR is induced when any of the three components is omitted (Fig. 1a).

Transient expression of Cf-2, SpRcr3, and Avr2 by agroinfiltration in tobacco (Nicotiana tabacum cv. Petite Havana SR1) also induces HR (Fig. 1b), which is also absent when any of the three components is omitted (Fig. 1b). These data show that HR signalling downstream of Cf-2/Rcr3 is conserved in Nicotiana species, and that these Nicotiana species do not already carry a functional Cf-2 or Rcr3. These data indicate that either Rcr3 is absent in Nicotiana species, consistent with the hypothesis that Rcr3 evolved only recently, or that the Nicotiana Rcr3 is not able to participate in Cf-2-dependent Avr2 recognition.

We next replaced each of the three components to study the specificity of the transient HR assay. We first replaced Avr2 with a previously described Avr2 mutant, which lacks the last six residues (Avr2Δ6) and which cannot inhibit Rcr3 nor trigger HR in tomato lines containing Cf-2[14]. Indeed, Avr2Δ6 does not trigger a HR when co-expressed with SpRcr3 and Cf-2 (Fig. 1c). By contrast, the only known natural variant of Avr2 carrying a leucine to valine (L43V) substitution (Avr2(L43V)[15], triggers HR upon co-expression with Cf-2 and SpRcr3 (Fig. 1c). Finally, Cf-2 is reported to recognize the nematode Globodera rostochiensis effector GrVAP1 in an Rcr3-dependent manner[12]. However, co-expression of GrVAP1 with Cf-2 and SpRcr3 in N. benthamiana by agroinfiltration did not trigger HR (Fig. 1d). We therefore omitted GrVAP1 from subsequent assays.

Likewise, we replaced SpRcr3 in our transient assay. We first replaced SpRcr3 with its closely related tomato paralog, SlPip1. SlPip1 does not trigger HR upon co-infiltration with Cf-2 and Avr2 (Fig. 1e), consistent with the fact that tomato lines that lack Rcr3 (Money Maker (MM)-Cf-2 rcr3-3) but still contain SlPip1 cannot trigger Cf-2-dependent Avr2 recognition[9]. Next, we investigated auto-necrosis induced by SlRcr3, the Rcr3 from cultivated tomato (S. lycopersicum). SlRcr3 induces Avr2-independent necrosis in tomato lines carrying Cf-2[6]. However,

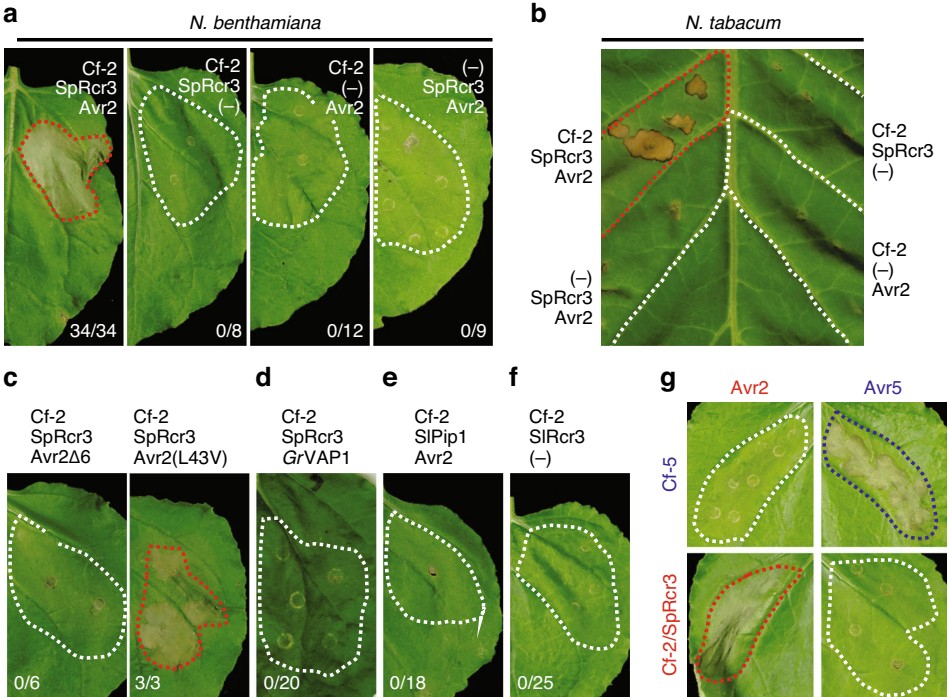

**Fig. 1 Transient co-expression of Rcr3 with Cf-2 and Avr2 triggers HR. a** Co-expression of Cf-2, *Sp*Rcr3, and Avr2 triggers a HR in *N. benthamiana*. Agrobacterium containing binary vectors for the expression of *Sp*Rcr3, Avr2 and a codon-optimized *Cf-2* were infiltrated in fully expanded *N. benthamiana* leaves. Omitting one of the three components results in a loss of HR. **b** Co-expression of *Sp*Rcr3 with Cf-2 and Avr2 triggers HR in tobacco. *Sp*Rcr3, Cf-2, and Avr2 were expressed in *N. tabacum* cv. *Petite Havana SR1* by agroinfiltration without the P19 silencing inhibitor. **c** HR is triggered by Avr2(L43V) but not by Avr2Δ6. *Sp*Rcr3, and Cf-2 were co-expressed with either an Avr2 mutant (Avr2Δ6) or with a natural variant of Avr2 containing the L43V substitution. **d** Transient co-expression of nematode effector *Gr*VAP1 with *Sp*Rcr3 and Cf-2 does not trigger a HR. **e** *Sl*Pip1 does not trigger HR when co-expressed with Cf-2 and Avr2. **f** The auto-necrosis-inducing *S. lycopersicum* allele of Rcr3 (*Sl*Rcr3) does not trigger a HR when co-expressed with Cf-2 in the absence of Avr2. **g** Transient expression of codon-optimized Cf-5 with Avr5 triggers a HR. **a–g** Agrobacterium containing different binary vectors were combined at a final $OD_{600} = 0.25$ each in a 1:1:1 ratio. The silencing inhibitor P19 was present in the backbone of all binary vectors except for those used in tobacco. All pictures taken at 5 dpi. Numbers indicate the number of leaves showing HR. Infiltrated sectors showing HR are encircled with a red dash line.

we were unable to trigger HR or necrosis upon transient expression or *Sl*Rcr3 with Cf-2 in the absence of Avr2 (Fig. 1f). This indicates that either our assay is not sensitive enough to detect this necrosis, or that an additional component from tomato is absent in this assay.

Finally, we replaced *Cf-2* in our transient assays. *Cf-5* is allelic to *Cf-2* and encodes an Cf-2-like RLP that provides resistance against *C. fulvum* carrying *Avr5*[16]. *Avr5* encodes a small cysteine-rich secreted protein that has no significant sequence homology to proteins with known functions[17]. Replacement of Cf-2 by Cf-5 did not trigger HR with Avr2 and *Sp*Rcr3, showing that Avr2/Rcr3-induced HR is *Cf-2*-specific (Fig. 1g). Co-expression of Avr5 with Cf-5 triggers HR, indicating that this Avr5/Cf-5-induced HR does not require Rcr3 (Fig. 1g). These data are consistent with previous observations that *Cf-5* does not require *Rcr3* for the recognition of *C. fulvum* races producing Avr5[18]. Finally, *Sp*Rcr3/Cf-2 does not trigger HR with Avr5 (Fig. 1g), consistent with the classic gene-for-gene model.

**Resurrected *Nb*Rcr3 is active but cannot trigger HR.** In the experiments described above, we observed that Cf-2-dependent Avr2 recognition in *N. benthamiana* requires co-expression with *Sp*Rcr3 (Fig. 1a). We therefore investigated if *N. benthamiana* lacks Rcr3, or whether *N. benthamiana* Rcr3 is unable to interact with either Avr2 or Cf-2. In the previously published *N. benthamiana* draft genome v1.0.1[19], we identified two *Rcr3* homologs (*Nb*Rcr3a and *Nb*Rcr3b) and one *Pip1* homolog (*Nb*Pip1, Supplementary Fig. 1). Phylogenetic analysis with tomato PLCPs demonstrates that *Nb*Rcr3a and *Nb*Rcr3b cluster within the

Rcr3 subclade and *Nb*Pip1 clusters within the Pip1 subclade (Supplementary Fig. 2), indicating that these Rcr3 and Pip1 orthologs, respectively. *Nb*Rcr3a, *Nb*Rcr3b, and *Nb*Pip1 were also identified in an independent genome assembly[20] and in the QLD accession[21] and no additional sequences with >80% nucleotide identity were identified with blast searches. Since *N. benthamiana* is alloploid, the two *Nb*Rcr3 paralogs might be homeologs derived from the two ancestral genomes.

*Nb*Rcr3a and *Nb*Pip1 reside on a single contig (Fig. 2a). This locus is syntenic with the *Rcr3/Pip1* locus in tomato, potato, and pepper[10] because this block also contains homologs of the *Pip1-flanking protease 1* (*Nb*Pfp1) and *Rcr3-flanking protease 1* (*Nb*Rfp1), as well as a lipase homolog found between *Rcr3* and *Pip1* in tomato (Fig. 2a). *Nb*Rcr3b is located on a different contig containing several lipase genes.

A closer look revealed two frame-shift mutations in *Nb*Rcr3a and one frame-shift mutation in *Nb*Rcr3b that would inactivate the encoded proteins (Fig. 2 and Supplementary Fig. 1). Interestingly, the frame-shift mutations in *Nb*Rcr3a in the LAB accession of *N. benthamiana* are different from those found in the NWA and NT accessions[21], indicating these mutations occurred independently and relatively recently in this species (Fig. 2b). We were unable to identify *Nb*Rcr3a in the published de novo transcriptomes of the *N. benthamiana* SA, WA, or QLD accessions[21], nor could we identify full-length *Nb*Rcr3b sequences from these accessions.

We corrected the frameshifts in *Nb*Rcr3a and expressed this engineered full-length *Nb*Rcr3a-1 allele by agroinfiltration. Activity-based protein profiling with the MV201 probe, a

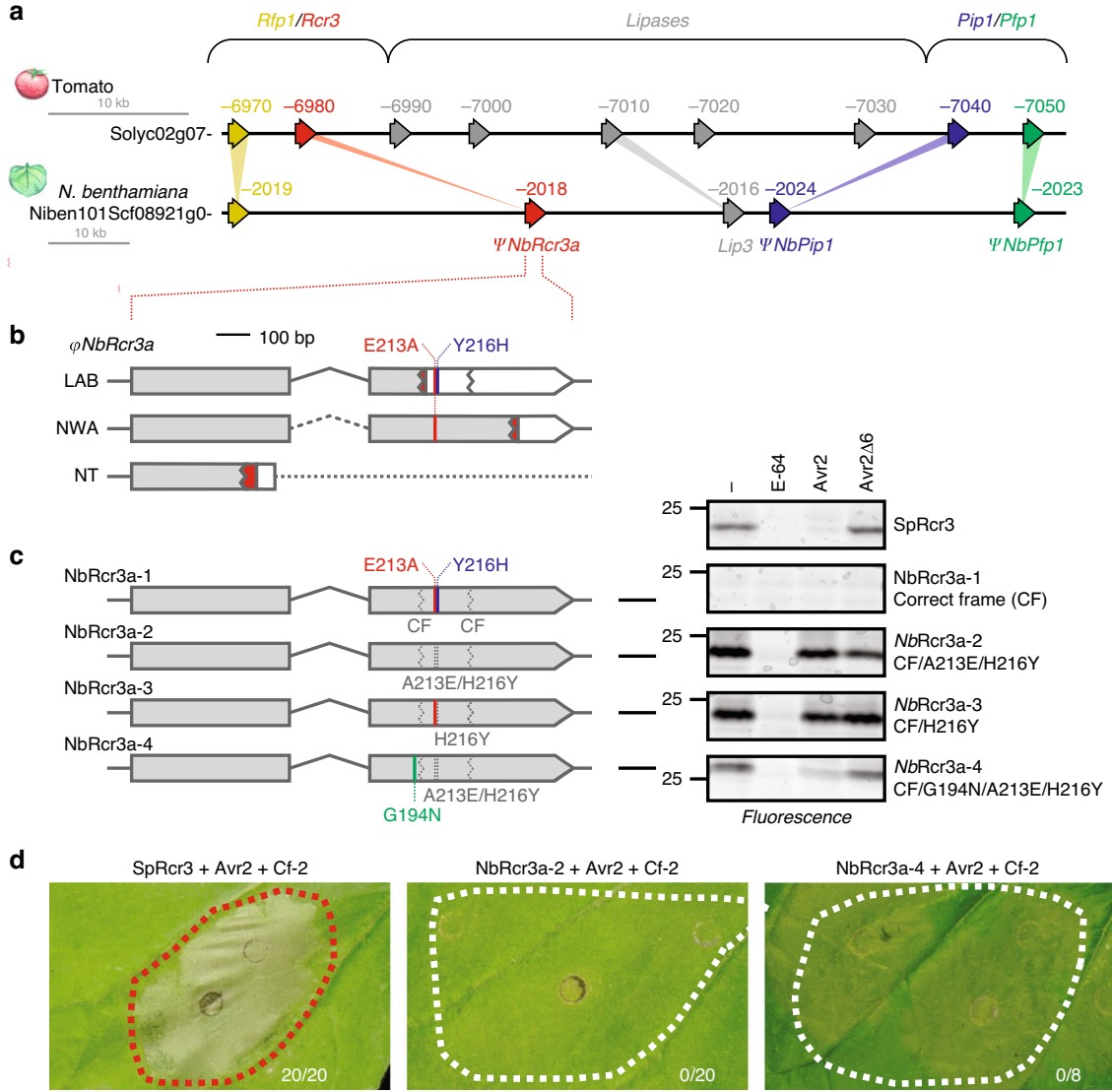

**Fig. 2 Resurrection of *NbRcr3a* null mutant establishes activity and Avr2 inhibition but not HR. a** Synteny between tomato and *N. benthamiana Rcr3/Pip1* loci. **b** Schematic representation of mutations in *NbRcr3a* in different *N. benthamiana* accessions. Dotted line indicates unknown sequence; dashed line indicates presumed intron sequence; red indicates a different ORF after indels, white indicates the untranslated region overlapping with the *Rcr3* ORF in a different frame. Two additional SNPs causing E213A and Y216H substitutions are indicated in red and blue, respectively. **c** Resurrection of *NbRcr3a* activity and Avr2 inhibition. Tomato *Sp*Rcr3 and *Nb*Rcr3a mutants 1–4 (left) were produced by agroinfiltration of *N. benthamiana* leaves. Apoplastic fluids were isolated at 2 dpi and preincubated for 45 min with and without 100 μM E-64, 2 μM Avr2 or Avr2Δ6, and labelled for 3 h with 2 μM MV201. Samples were separated on SDS-PAGE gels and scanned for fluorescence. **d** Transient co-expression of resurrected *Nb*Rcr3a-2 or -4 with Cf-2 and Avr2 does not trigger a HR in *N. benthamiana*. Pictures were taken at 5 dpi. Numbers indicate the number of leaves showing HR.

fluorescent derivative of the general PLCP inhibitor E-64[22], was used to test the activity of *Nb*Rcr3a-1. However, *Nb*Rcr3a-1 is not an active protease, in contrast to tomato Rcr3 (Fig. 2c). Upon aligning the *Nb*Rcr3a-1 protein with all Arabidopsis and tomato PLCPs, we realized there were two additional substitutions of otherwise highly conserved residues: a glutamic acid into alanine substitution (E213A) and a tyrosine into histidine substitution (Y216H). Correcting both E213A and H216Y restores protease activity (Fig. 2c, *Nb*Rcr3a-2), and the H216Y substitution is sufficient to restore activity (Fig. 2c, *Nb*Rcr3a-3). The E213A substitution is present in *N. benthamiana* accessions LAB and NWA (Fig. 2b) suggesting that this substitution may have occurred first, with subsequent frame-shift mutations and non-synonymous substitutions occurring later in different *N. benthamiana* populations.

Although *Nb*Rcr3a-2 is an active protease, it is not inhibited by Avr2 (Fig. 2c). Consistent with its lack of Avr2 inhibition, NbRcr3a-2 cannot trigger HR when co-expressed with Cf-2 and Avr2 in *N. benthamiana* (Fig. 2d). We suspected this is because *Nb*Rcr3a-2 carries a glycine at position 194, as a naturally occurring allele of tomato Rcr3 carrying a N194D substitution was not inhibited by Avr2[9]. Indeed, the *Nb*Rcr3a-4 mutant carrying G194N is efficiently inhibited by Avr2 (Fig. 2c), confirming that N194 enhances the interaction between Rcr3 and Avr2. However, co-expression of *Nb*Rcr3a-4 with Cf-2 and Avr2 does not trigger HR in *N. benthamiana* (Fig. 2d), indicating that *Nb*Rcr3a-4 is unable to signal via Cf-2. Thus, *Nb*Rcr3a engineering resulted in a protease that is active and can be inhibited by Avr2, but could not induce HR upon co-expression with Avr2 and Cf-2.

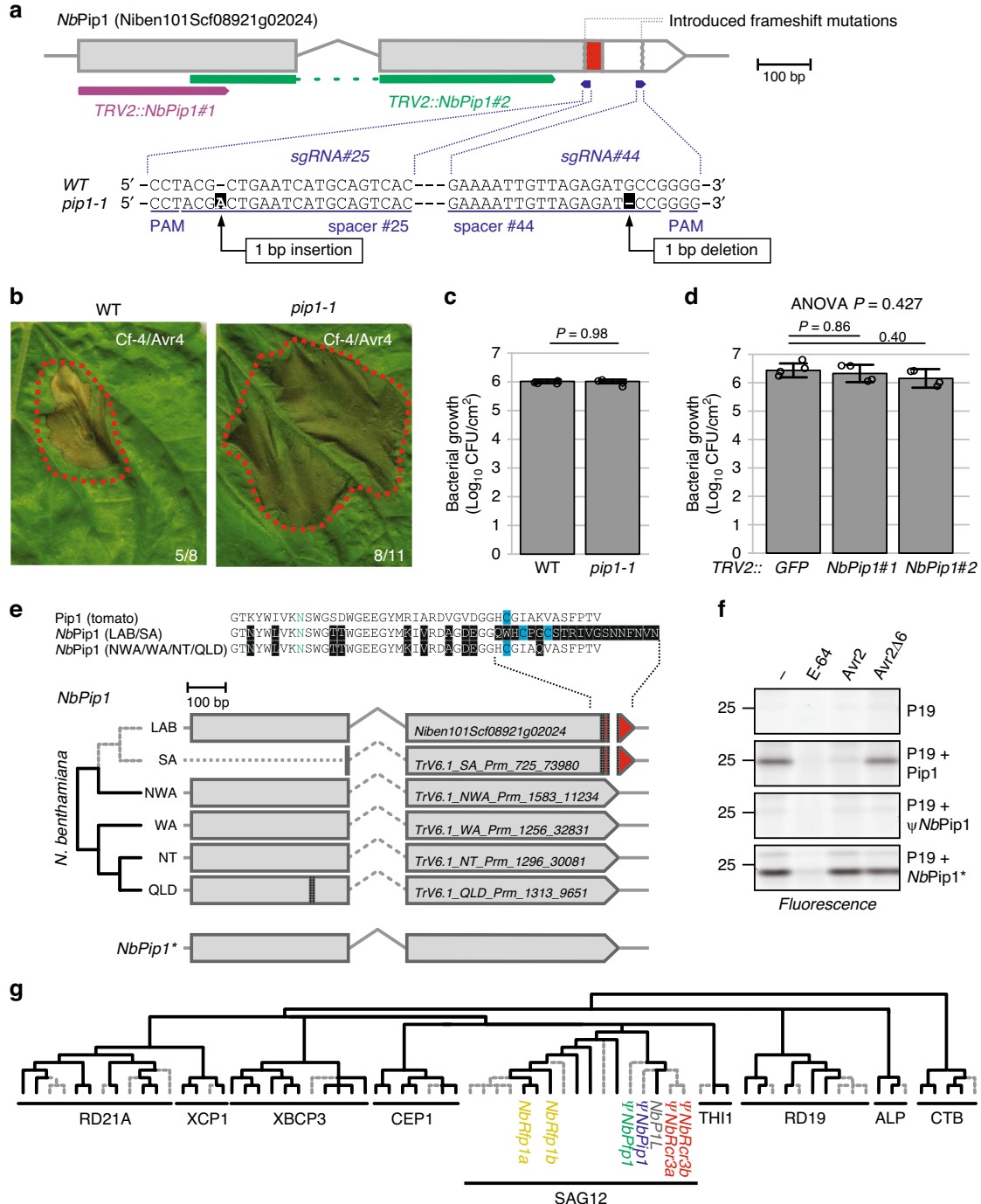

**Pip1 is a pseudogene in N. benthamiana**. We next studied *NbPip1*, which is a paralog of *NbRcr3a*. *Pip1* is highly conserved in Solanaceae (Supplementary Fig. 2). *Pip1* acts as an important immune protease in tomato, providing basal resistance against bacterial (*Pseudomonas syringae*), fungal (*Cladosporium fulvum*), and oomycete (*Phytophthora infestans*) pathogens[10]. Pip1 activity is also strongly increased prior to Cf-4/Avr4-dependent HR[23], and silencing of *Pip1* in *N. benthamiana* blocked HR induced by Cf-4/Avr4[24]. We therefore tested the role of *NbPip1* in immunity. We used CRISPR/Cas9 genome editing[25] to generate a mutant line carrying two frame-shift mutations in *NbPip1* (*pip1-1* line) (Fig. 3a). Surprisingly, the *N. benthamiana pip1-1* line was still able to trigger HR upon co-expression of Cf-4 and Avr4 (Fig. 3b), in contrast to earlier studies[23]. In addition, the *pip1-1* line is not

more susceptible to *Pseudomonas syringae* pv. *tomato* DC3000 Δ*hopQ1-1* (*Pto*DC3000Δ*hQ*) (Fig. 3c, d), unlike antisense *Pip1* tomato lines[10]. Also *N. benthamiana* plants silenced for *NbPip1* using virus-induced gene silencing (VIGS), are not more susceptible to *Pto*DC3000Δ*hQ* (Fig. 3d).

We therefore took a closer look at the *NbPip1* gene sequence. We realized that two deletions of 2 bp and 24 bp in the 3′ end of the *NbPip1* gene cause a frame-shift that replaces the last 13 amino acids with a different sequence of 20 amino acids (Fig. 3e). The 13 amino acid region contains an important cysteine required to make a disulphide-bridge critical for PLCP function[22]. Instead, the 20 amino acid replacing sequence contains two cysteines, but these residues are not in the same position and sequence context as the cysteine in the original sequence, and are

**Fig. 3 NbPip1 is a pseudogene, not required for immunity or HR signalling. a** Schematic representation of the *NbPip1* gene model. The CRISPR/Cas9 frame-shift mutant of *NbPip1* (*pip1-1*) carries a 1 bp deletion leading to an alternative open reading frame (ORF, red) and an untranslated ORF (white). Two fragments of *NbPip1* used for VIGS are indicated below the sequence. **b** Avr4/Cf-4-triggered HR does not require *NbPip1*. Cf-4 and Avr4 were transiently co-expressed by agroinfiltration of both WT and *pip1-1* mutant plants and pictures were taken 5 dpi. Numbers indicate the number of leaves showing HR. **c** The *pip1-1* mutant line is not more susceptible to *P. syringae* pv. *tomato* DC3000 *ΔhopQ1-1* (*Pto*DC3000ΔhQ). Plants were infiltrated with $10^5$ CFU/ml *Pto*DC3000ΔhQ and bacterial growth was measured by colony counting at 3 dpi. Error bars represent STDEV of $n = 4$ biological replicates. Probability values were calculated with Students t-test. **d** *Nb*Pip1 depletion by VIGS does not affect susceptibility to *Pto*DC3000ΔhQ. Young *N. benthamiana* plants were inoculated with *TRV::GFP*, *TRV::NbPip1#1*, or *TRV2::NbPip1#2* and infiltrated with $10^5$ CFU/ml *Pto*DC3000ΔhQ and bacterial growth was measured by colony counting at 3 dpi. Error bars represent STDEV of $n = 4$ biological replicates. Probability values were calculated with ANOVA using the Tukey's post-hoc test. **e** *Nb*Pip1 is a pseudogene in LAB and SA accessions. Two small deletions of 2 and 24 bp in the 3′end of *NbPip1* results in an altered C-terminus of the encoded protein (red). Dotted line indicates unknown sequence; dashed line indicates putative intron. Protein sequences contain polymorphic residues (black), catalytic Asn (green) and Cys residues (blue). **f** *Nb*Pip1 can be resurrected into an active protease by correcting the 2 and 24 bp deletions. Pip1, φ*Nb*Pip1, and an *Nb*Pip1 mutant with a corrected 3′ end (*Nb*Pip1*) were co-expressed with silencing inhibitor P19 by agroinfiltration. Apoplastic fluids were isolated at 4 dpi, preincubated for 45 min with or without 100 μM E-64, 2 μM Avr2 or Avr2Δ6, or DMSO, and labelled for 3 h with 2 μM MV201. Samples were separated on SDS-PAGE gels, scanned for fluorescence. **g** Pseudogenisation is common amongst PLCP subfamilies in *N. benthamiana*. The evolutionary history of the *N. benthamiana PLCP* gene family was inferred using the Maximum Likelihood method based on the Whelan and Goldman model[66]. The bootstrap consensus tree inferred from 1000 replicates is taken to represent the evolutionary history of the taxa analysed[67]. Putative inactive PLCPs are indicated with gray dotted lines. PLCP clades were named according to[22]. See Supplementary Fig. 3 for all names.

therefore probably unable to form a disulphide bridge. These data are consistent with the absence of peptides of *Nb*Pip1 in previously published proteomic datasets[26,27], indicating that *Nb*Pip1 protein does not accumulate.

The 2 bp and 24 bp deletions in *Nb*Pip1 must be relatively recent in the *N. benthamiana* species, as they are only present in the closely related LAB and SA accessions, but not in the more distantly related NWA, WA, NT, and QLD accessions (Fig. 3e), indicating that *Nb*Pip1 is an active protease in other accessions. Activity-based protein profiling shows that *Nb*Pip1 from the LAB accession is indeed not an active PLCP when produced by agroinfiltration (Fig. 3f). Correction of the two 3′ deletions results in an active *Nb*Pip1 (*Nb*Pip1*), but this protease is not inhibited by Avr2, unlike tomato Pip1 (Fig. 3f).

Because both *Rcr3* and *Pip1* are inactivated in *N. benthamiana*, we analysed the *PLCP* gene family in the *N. benthamiana* draft genome for additional inactivated genes. We manually curated the gene models in the *N. benthamiana* v1.0.1 draft genome assembly[19]. Genes with potentially inactivating frame-shift mutations and partial genes were cross-validated in other *N. benthamiana* genome assemblies[20]. This resulted in the identification of 71 *PLCP* genes, twice the number of most plants, consistent with the alloploid genome of *N. benthamiana*. However, 26 of these 71 PLCP genes contain mutations that would inactivate the encoded protein, but at least one functional homeolog remains for each of the nine PLCP subfamilies (Fig. 3g, Supplementary Fig. 3, Supplementary Fig. 4, Supplementary Data 1). This pattern of inactivation was similar to what we previously found for the *N. benthamiana* subtilisin gene family, where for many protein-encoding genes, the corresponding homeolog is inactivated[27]. Homeolog inactivation may be caused by the contraction of the functional genome after the polyploidization event approximately 6 million years ago (6Mya)[28].

**Avr2 inhibition and HR induction by Rcr3 homologs.** Having established how functionally diverged *Nb*Rcr3 and *Nb*Pip1 are from their tomato orthologs, we investigated Rcr3 homologs from other solanaceous plant species for which genomic data are available. We identified one or two *Rcr3* homologs in every analysed Solanaceae species. To ensure these are true *Rcr3* homologs, we also identified homologs of *Pip1*, as well as the flanking paralogs *Pfp1* and *Rfp1*. Similar to *Rcr3*, all analysed solanaceous genomes contain one or two homologs of *Pip1*, *Pfp1*, and *Rfp1* (Supplementary Data 1).

Phylogenetic analysis demonstrates that these cluster in distinct subclades, demonstrating that we identified true Rcr3 homologs (Supplementary Fig. 4). The *Rcr3*, *Pip1*, and *Pfp1* subclades cluster together in one subclade of the SAG12 PLCP subfamily and all reside on one genomic locus (Supplementary Fig. 4). Another *Pip1*-like (P1L) PLCP is also conserved in Solanaceae and phylogenetically clusters together with *Rcr3*, *Pip1*, and *Pfp1* but resides on a different genomic locus (Supplementary Fig. 2, Supplementary Fig. 4). *Rfp1* is more distantly related to *Rcr3*, *Pip1*, and *Pfp1*, and clusters together with SAG12-like PLCP-encoding genes that are located on different chromosomes (Supplementary Fig. 4). Since *Rcr3* and *Pip1* are present in all Solanaceae species, but not outside of this plant family (Supplementary Fig. 2), we conclude that *Rcr3* and *Pip1* evolved >50Mya, when the Solanaceae family diverged from other plant families[29].

We next analysed the Rcr3 homologs in greater detail. Phylogenetic relationships of the Rcr3 sequences are consistent with phylogenetic relations of the solanaceous plant species (Fig. 4a). Rcr3 sequences of tomato cluster together, as do those of potato and eggplant, and these three species are all grouped together in one *Solanum* cluster. Likewise, *Capsicum* (pepper) Rcr3 homologs cluster together, as well as *Nicotiana* Rcr3 homologs. Alignment of the polymorphic residues in the protease domain reveals that several sites are highly polymorphic even within species (Fig. 4b, red residues), whereas other sites seem to be conserved within species but polymorphic between species (Fig. 4b, gray residues). ConSurf prediction[30] indicates that the highly polymorphic residues are exposed to the surface on the protein model of the Rcr3 protease domain (Fig. 4c). The catalytic site (yellow) and the substrate-binding groove are well conserved. Most polymorphic residues locate around the substrate-binding groove as a 'ring of fire', and many of these residues are also polymorphic within both tomato and tobacco species. Only a few polymorphic sites are located on the back of the protein.

We cloned 13 Rcr3 homologs from potato, eggplant, pepper, petunia, and tobacco and transiently expressed them by agroinfiltration of *N. benthamiana*. MV201 labelling experiments showed that all these Rcr3 homologs are active proteases (Fig. 5a), making *N. benthamiana* the exception having no active Rcr3 protease. Furthermore, all Rcr3 homologs except for the *Nicotiana* Rcr3 homologs can be inhibited by Avr2, although Avr2 inhibition is less strong in *Capsicum* and *Petunia* Rcr3 (Fig. 5a). The *Nicotiana* Rcr3 homologs carry a lysine at position

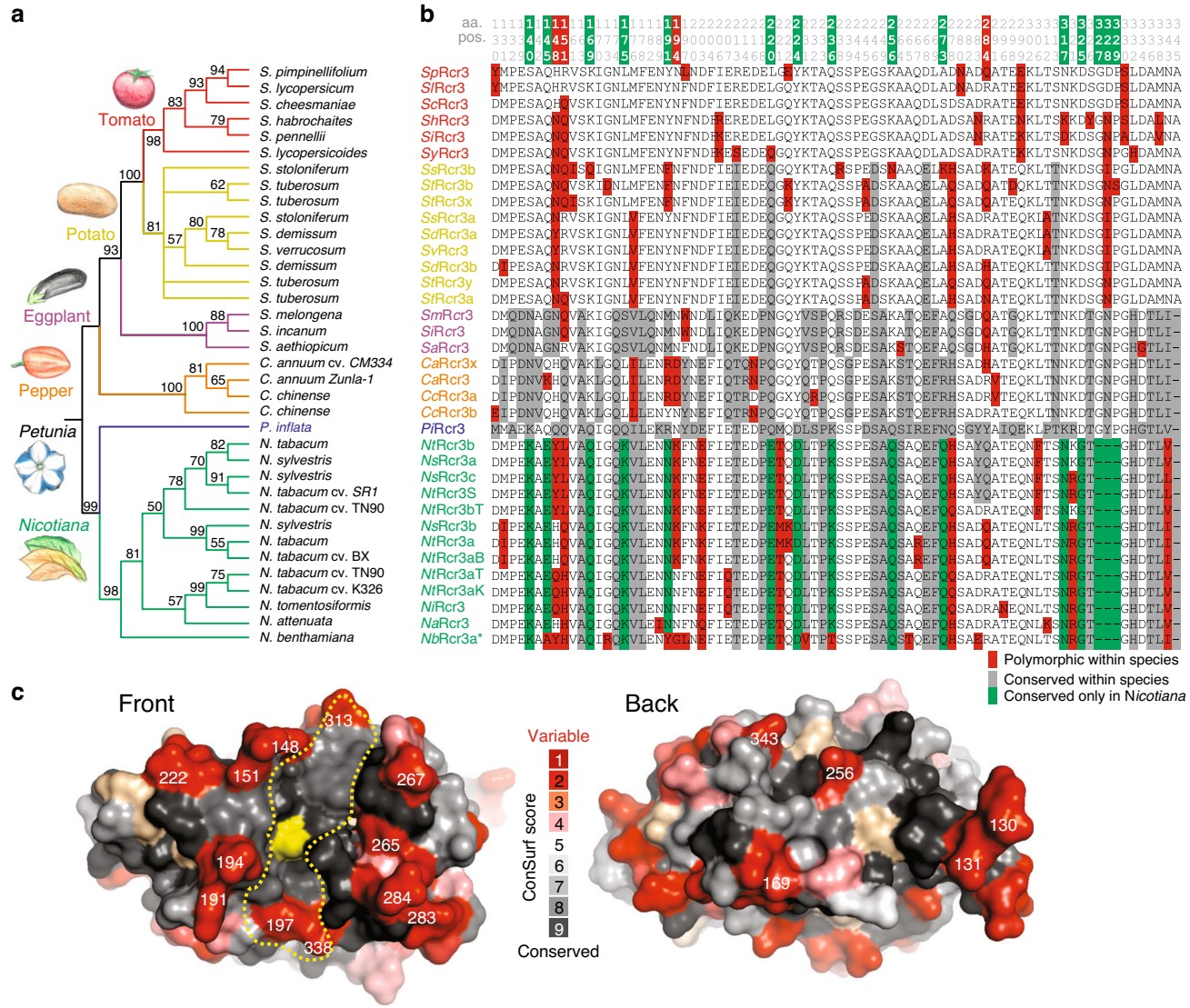

**Fig. 4 Rcr3 is highly polymorphic in Solanaceae. a** The evolutionary history of Rcr3 was inferred using the Maximum Likelihood method based on the Whelan and Goldman model. The bootstrap consensus tree inferred from 1000 replicates is taken to represent the evolutionary history. Shown are percentage bootstrap values. **b** Summary of variant residues in the protease domain. Columns with conserved residues or a single substitution are not shown. Highlighted are residues that are polymorphic within a genus (red) polymorphic within solanaceous plants but putatively fixed within a genus (gray), and putatively fixed only within *Nicotiana* (green). The residue position is indicated on top (e.g., first residue is pos. 130) with the residues relevant for Avr2 binding highlighted in red and residues associated with HR in green. *, resurrected *Nb*Rcr3a* is not included in the annotation of *Nicotiana*-specific sites (green). **c** Position of polymorphic residues in the 3D model of Rcr3. *Sl*Rcr3 was modelled on 1s4v and amino acid polymorphism were analysed by ConSurf and scores were used to color the structure using PyMol. Highlighted are the catalytic Cys residue (yellow) and substrate-binding groove (yellow dash line), as well as some of the most polymorphic residues (white numbers).

194, which may prevent inhibition by Avr2[9]. Substituting this lysine by asparagine (K194N) indeed results in *Nicotiana* Rcr3 mutants that are sensitive to Avr2 inhibition (Supplementary Fig. 5).

All tested tomato and potato Rcr3 homologs trigger HR when co-expressed with Cf-2 and Avr2 (Fig. 5b). By contrast, the tested eggplant, *Capsicum*, *Nicotiana,* and *Petunia* Rcr3 homologs do not trigger HR when co-expressed with Cf-2 and Avr2 (Fig. 3c). Also the *Nicotiana* Rcr3 K194N mutants do not trigger HR when co-expressed with Avr2 and Cf-2 (Supplementary Fig. 5), indicating that Rcr3 homologs outside the *Solanum* genus are probably unable to signal with Cf-2. Thus, the ability of Rcr3 homologs to participate in Cf-2-dependent Avr2 recognition correlates with the evolutionary distance to tomato, which is consistent with Rcr3/Cf-2 coevolution in tomato. It seems likely

that the ability to trigger HR resides in species-specific sites in Rcr3 (Fig. 4b, green).

**Four variant residues in Rcr3 determine sensitivity to Avr2.** To investigate if tomato Rcr3 evolved maximal affinity to interact with Avr2, we examined the role of natural variation in Rcr3 homologs from different species. Avr2 inhibits most Rcr3 homologs, and N194 is an important determinant of this interaction. However, we realized that N194 is not the only polymorphic residue required for Avr2 inhibition. Pepper Rcr3, for instance, has an aspartic acid residue at this position (D194), but is still efficiently inhibited by Avr2 (Fig. 5a). Similarly, Pip1 has a glutamic acid at this position (E194), but is also efficiently inhibited by Avr2[9,31]. This indicates that there are variant residues in Rcr3 that may compensate for the absence of N194.

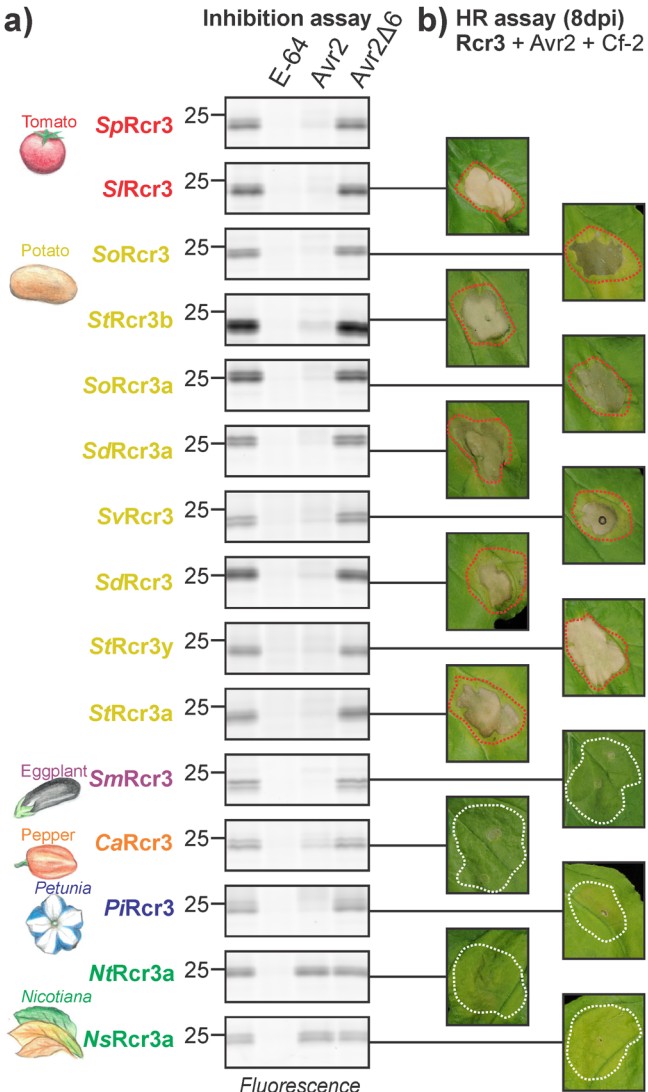

**Fig. 5 Rcr3 variants are diverging on Avr2 inhibition and HR signalling.**
**a** Activity profiling of Rcr3 homologs. Rcr3 homologs were co-expressed with the silencing inhibitor P19 in *N. benthamiana* leaves by agroinfiltration. Apoplastic fluids were isolated at 2 dpi, preincubated for 45 min with or without 100 μM E-64, 4 μM Avr2 or Avr2Δ6, or DMSO, and labelled for 3 h with 2 μM MV201. Samples were separated on SDS-PAGE gels and scanned for fluorescence. **b** Tomato and potato Rcr3 homologs can trigger a HR, while the pepper, eggplant, *Nicotiana* and *Petunia* Rcr3 homologs do not trigger responses. Rcr3 homologs were co-expressed with Avr2 and Cf-2 by mixing Agrobacterium cultures (OD$_{600}$ = 0.25 each) in a 1:1:1 ratio. Pictures were taken at 8 dpi.

It was previously noticed that some tomato *Rcr3* alleles carrying the N194D substitution can still trigger HR in the presence of *Cf-2* and Avr2[32]. This ability to trigger HR was associated with the presence of glutamine instead of arginine at position 151 (R151Q)[32]. Indeed, *Ca*Rcr3 also carries Q151, while *Sl*Pip1 carries V151. To test this association, we expressed N194D, R151Q/N194D, and R151V/N194D mutants of *Sp*Rcr3. *Sp*Rcr3(N194D) is not inhibited by Avr2 at the tested concentrations (Fig. 6a), while *Sp*Rcr3(R151Q/N194D) and *Sp*Rcr3(R151V/N194D) are inhibited by Avr2, confirming a role for R151 in the interaction with Avr2 (Fig. 6a). Similarly, the V151R substitution in *Sl*Pip1 reduces inhibition by Avr2, indicating that R151 acts as a repulsive residue at the Rcr3-Avr2 interface (Fig. 6b).

However, in contrast to *Sp*Rcr3(N194D), the N194D substitution mutant of the *S. lycopersicum* allele (*Sl*Rcr3(N194D)) was still inhibited by Avr2 while both contain R151 (Fig. 6c). *Sp*Rcr3 and *Sl*Rcr3 differ by only six amino acid residues in the mature protease. One of these polymorphisms, Q284R, is specific to *Sp*Rcr3 when compared to other tomato Rcr3 homologs[32]. *Sl*Pip1 also carries R284. Since this residue resides close to the substrate-binding groove, similar to R151 and N194 (Fig. 6e), we investigated its role in the Avr2 interaction. Indeed, *Sl*Pip1 (R284Q) (Fig. 6b) and *Sl*Rcr3(N194D/R284Q) (Fig. 6c) are less efficiently inhibited by Avr2. By contrast, *Sp*Rcr3(N194D/Q284R) as well as *Sp*Rcr3(R151Q/N194D/Q284R) and *Sp*Rcr3(R151V/ N194D/Q284R) are more efficiently inhibited by Avr2 (Fig. 6a). These data confirm a role for the Q284R polymorphism in Avr2 inhibition and indicates that R284 promotes the interaction with Avr2.

Finally, H148D in *Sl*Rcr3 increases inhibition by the *Phytophthora mirabilis* effector *Pm*EPIC1[33]. *Sl*Pip1 also contains an aspartic acid at this position (D148). We therefore tested whether this residue is also involved in the interaction with Avr2. Surprisingly, both *Sp*Rcr3(H148D) and *Sl*Pip1(D148H) mutants did not accumulate as active proteases (Fig. 6a, b) for unknown reasons. However, combination of H148D with either R151V or Q284R substitutions in *Sp*Rcr3 restored protease activity (Fig. 6a). Similarly, activity of the *Sl*Pip1(D148H) mutant was restored by the V151R substitution, and a slightly increased activity could be restored by the R284Q substitution (Fig. 6b). Introduction of the H148D substitution into the *Sp*Rcr3(R151V/N194D) and *Sp*Rcr3 (N194D/Q284R) mutants further enhanced the inhibition by Avr2 (Fig. 6a). Similarly, D148H mutation in *Sl*Pip1 results in reduced inhibition by Avr2, which is additive to the V151R and R284Q mutations (Fig. 6b). To confirm our findings, we made similar mutants in *Nb*Rcr3a-2. Mutating Y148D, H151V, or G194N independently resulted in increased Avr2 inhibition in *Nb*Rcr3a-2, and combining these residues had an additive effect (Fig. 6d).

The four residues at positions 148, 151, 194, and 284 surround the substrate-binding groove and indicate a possible foot print of the interaction interface with Avr2 (Fig. 6e). The presence of specific residues at these positions may predict whether PLCPs are sensitive to Avr2 inhibition. To test this prediction, we made mutants of the unrelated tomato PLCPs C14 and Cyp3, which cannot be inhibited by Avr2 (Supplementary Fig. 6)[9,31]. The D194N or A284R substitutions in C14, or the D284R substitution in Cyp3 results in increased sensitivity to Avr2 inhibition (Supplementary Fig. 6), confirming that these residues are crucial for interaction with Avr2. Interestingly, *Sp*Rcr3, which co-evolved with Cf-2, contains only one of the four residues required for optimal inhibition by Avr2 (Fig. 6f).

We hypothesize that Cf-2-dependent recognition of Avr2 could dependent on the strength of the Rcr3-Avr2 interaction. To test this hypothesis, we generated mutant versions of pepper Rcr3 (*Ca*Rcr3) to increase its interaction with Avr2. *Ca*Rcr3 is weakly inhibited by Avr2, even though it contains D194, likely because of compensation by Q151 and R284 (Fig. 5a). *Ca*Rcr3 does not trigger HR when co-expressed with Cf-2 and Avr2 (Fig. 5b and Fig. 6g). However, engineered *Ca*Rcr3 carrying H148D or D194N trigger HR upon co-expression with Avr2 and Cf-2, whereas stronger HR was triggered by the mutant carrying both substitutions (Fig. 6g). This indicates that HR signalling can be instated in *Ca*Rcr3 by introducing residues that can enhance the Rcr3-Avr2 interaction. However, a strong interaction with Avr2 is not sufficient for HR signalling, as the *Nb*Rcr3a-2 mutants with optimal residues for Avr2 interaction were unable to trigger HR upon co-expression with Cf-2 and Avr2 (Fig. 6h), indicating that residues other than those

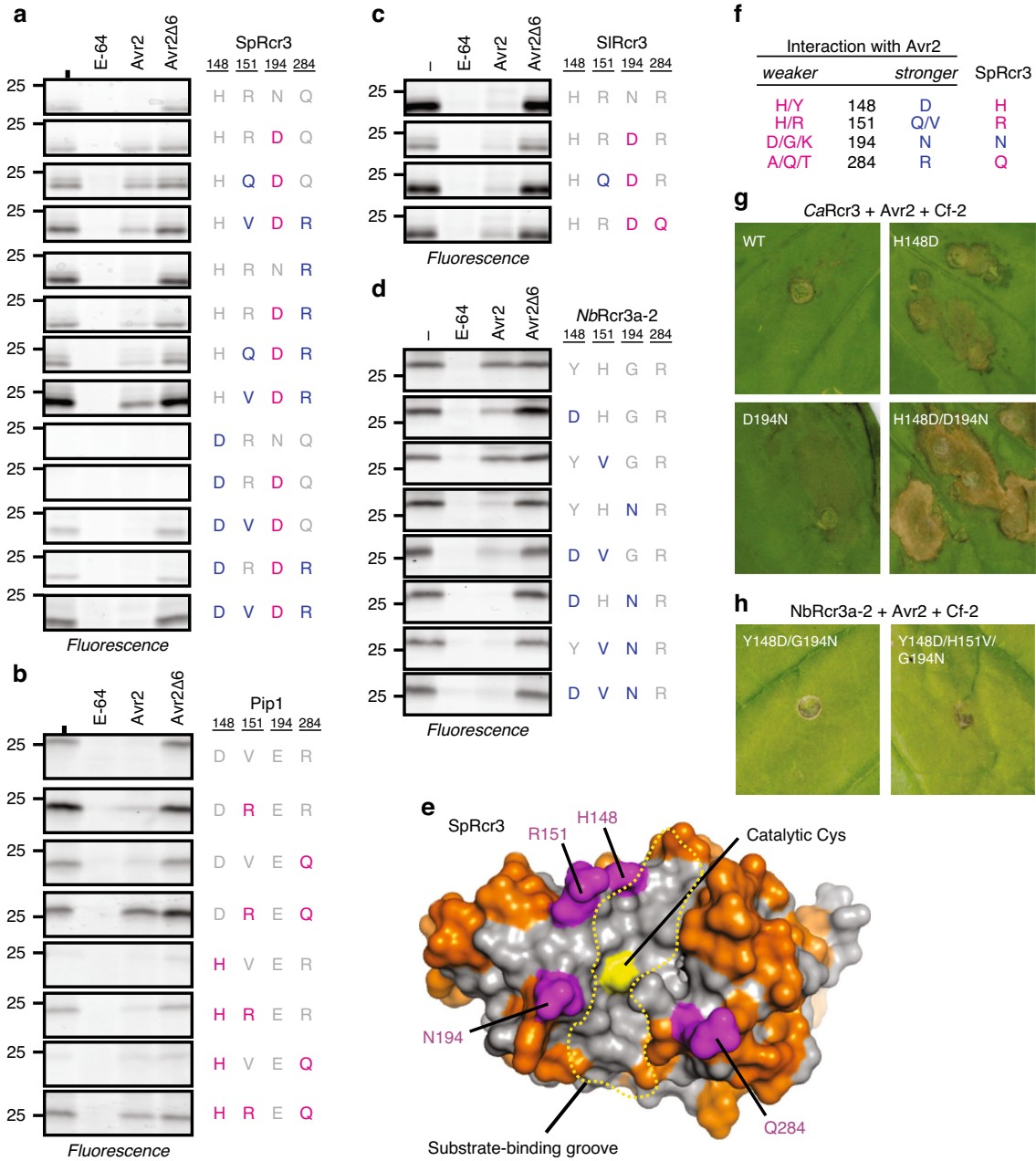

**Fig. 6 PLCP inhibition by Avr2 is determined by four residues.** Substitution mutants of *Sp*Rcr3 (**a**), *Sl*Pip1 (**b**), *Sl*Rcr3 (**c**), and *Nb*Rcr3a-2 (**d**) were produced by agroinfiltration of *N. benthamiana* leaves. Apoplastic fluids were isolated at 2 dpi, preincubated for 45 min with or without 100 μM E-64, 2 μM Avr2 or Avr2Δ6, or DMSO, and labelled for 3 h with 2 μM MV201. Samples were separated on SDS-PAGE gels and scanned for fluorescence. **e** H148, R151, N194, and Q284 residues (magenta) locate around the substrate-binding groove in Rcr3. Homology model of *Sl*Rcr3 based on 1s4v[10,68]. Shown are the substrate-binding groove (dotted line), catalytic cysteine (yellow), and polymorphic residues between *Sp*Rcr3 and *Sl*Pip1 (orange). **f** Summary of the effect of variant residues on inhibition by Avr2, compared to residues present in *Sp*Rcr3. **g** Pepper Rcr3 (*Ca*Rcr3) substitutions enhance Avr2-triggered HR. **h** *Nb*Rcr3a-2 that carry substitutions enhancing inhibition by Avr2 without triggering HR. Mutant *Ca*Rcr3 or *Nb*Rcr3a-2 were transiently co-expressed with Cf-2 and Avr2 by agroinfiltration and pictures taken at 5 dpi.

involved in the interaction with Avr2 are also required for Cf-2-dependent recognition of Avr2.

**Cf-2 belongs to a specific clade of the *Hcr2* gene family.** We next investigated when Cf-2 evolved. In contrast to *Rcr3*, *Cf-2* has only recently evolved by intragenic recombination[4]. This intragenic recombination results in a highly repetitive gene structure evident at nucleotide and amino acid level[4]. Both *Cf-2* and *Cf-5* belong to the *Hcr2* (for homologous to *Cladosporium* resistance

gene *Cf-2*) gene family[34]. To investigate the evolutionary origins of *Cf-2* and *Cf-5*, we constructed a phylogenetic tree of Hcr2s based on the amino acids encoded by the region corresponding to the conserved 5' and 3' region (Fig. 7a). We identified genes containing both these conserved 5' and 3' sequences in all Solanaceae species except *Petunia*.

The Solanaceae *Hcr2* gene family is divided in three well-supported clades (Fig. 7a). Clade-1 contains *Cf-2* and *Cf-5* and is restricted to *Solanum* genes. This Clade-1 can be further subdivided into two previously described Hcr2 clades, which

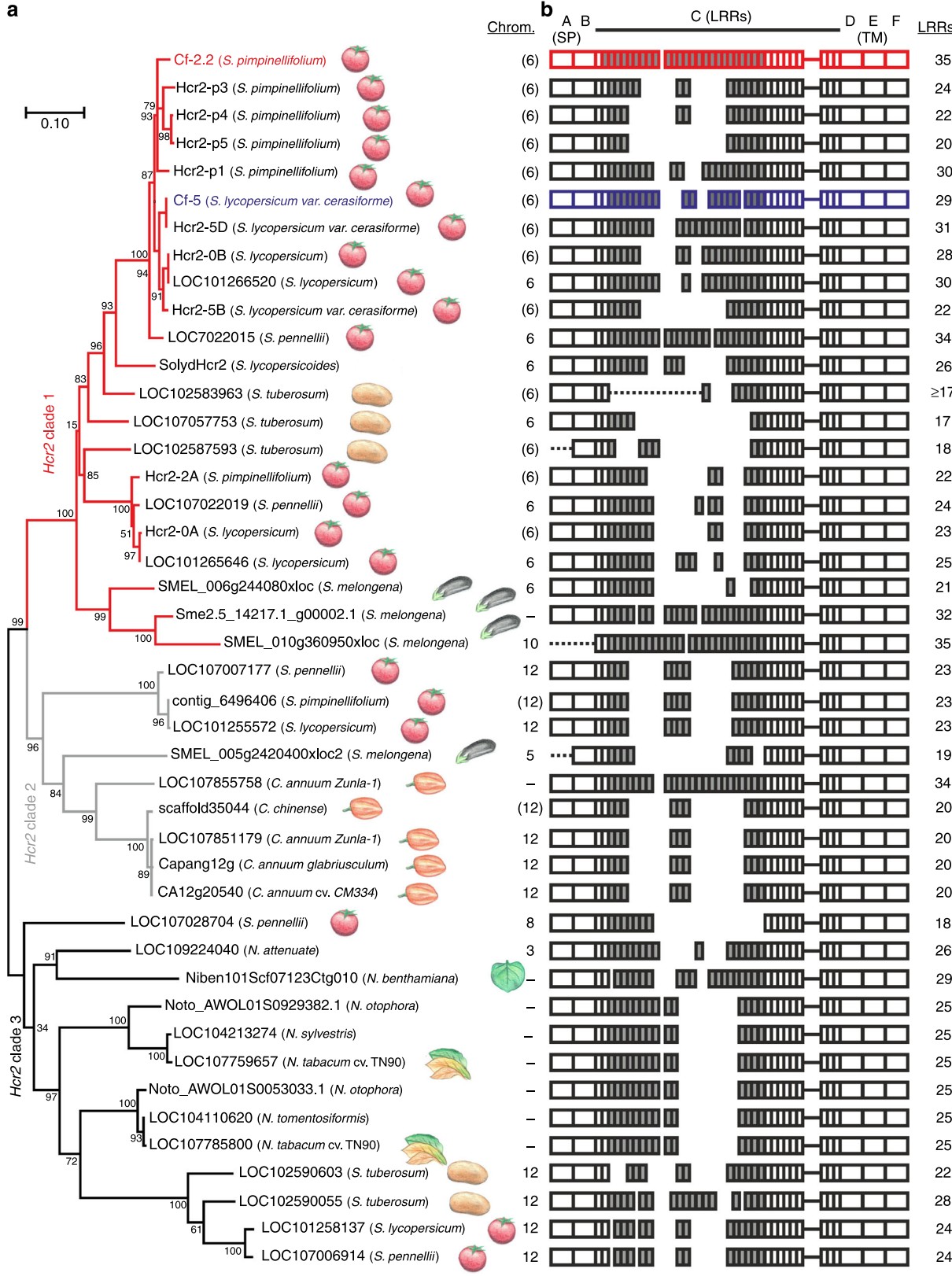

evolved separately on chromosome 6 and are represented by the allelic *Cf-2*/*Cf-5* genes and the paralogous *Hcr2-2A*, respectively[16], as well as an eggplant-specific subclade (Fig. 7a). Clade-2 contains *Capsicum* and *Solanum* genes, whereas Clade-3 contains *Solanum* and *Nicotiana* genes. This phylogeny indicates that *Hcr2s* from all three clades are present in *Solanum* (Fig. 7a). Clade-1 is evolutionary recent, as it only contains *Solanum* genes, including *Cf-2* and *Cf-5*. By contrast, Clade-3 contains homologs from across Solanaceae and may thus be the evolutionary more ancestral clade.

**Fig. 7 Cf-2 evolved in a tomato-specific subclade of the Hcr2 gene family. a** Phylogenetic tree of the *Hcr2* gene family based on the conserved *N*-terminal and *C*-terminal regions (corresponding to residues 1-138 and 713-1112 of Cf-2). Approximately Maximum Likelihood tree was generated using FastTreeMP, based on the LG model[69]. The bootstrap consensus tree inferred from 1000 replicates is taken to represent the evolutionary history of the taxa analysed. Shown are percentage bootstrap values. *Hcr2* family members with known recognition specificity are indicated in bold, the chromosome on which these genes reside is indicated, expected chromosome numbers are indicated in parentheses. Scale bar, amino acid substitutions per site. **b** Amino acid alignment comparison to Cf-2 using MUSCLE. LRRs of domain C were predicted using LRRfinder[70]. Numbers indicate total number of LRRs in domain C. Shown are LRR region evolving by intragenic recombination (gray); Cf-2 (red) and Cf-5 (blue); unknown sequences (dotted lines). A, signal peptide; B, N-terminal LRR flanking domain; C, LRR region; D, juxtamembrane region; E, transmembrane domain; F, cytoplasmic tail.

*Hcr2* homologs evolve by intragenic recombination, generating genes encoding for varying numbers of 24 amino acid LRRs (Fig. 7b). Some of these LRRs may be identical between different Hcr2 homologs, but occur in different positions of the protein, indicating that sequence exchange occurs in the variable LRR region between *Hcr2* homologs on different genomic loci. Interestingly, we also identified a highly conserved gene family in Solanaceous plants that contains the second half of *Cf-2*, but has a different N-terminal half, indicating that the *Hcr2* family evolved from this group of RLPs by replacing the N-terminal half of the protein by different set of recombining LRRs (Supplementary Fig. 7).

To investigate whether the pattern of evolution seen for the *Hcr2* gene family is unique to *Hcr2s*, we also investigated the *Hcr9* gene family in Solanaceae, which includes *R* genes *Cf-9*[35], *Cf-4*[36], *Hcr9-4E*[37], and *9DC*[38]. The *Hcr9* gene family has representatives in tobacco and tomato and other solanaceous plant species (Supplementary Fig. 8). In contrast to Hcr2 proteins, Hcr9 proteins have similar number of LRRs and missing LRRs in some family members are caused by deletions (Supplementary Fig. 8). Consequently, unlike *Hcr2* homologs, *Hcr9* homologs do not have the same repetitive gene structure at nucleotide level. Thus, while the *Hcr2* gene family evolves by intragenic recombination in the variable LRR-encoding domain, giving rise to highly repetitive regions encoding for differing numbers of LRRs, the *Hcr9* family evolves by diversifying selection within the LRR-encoding region. These data confirm that Cf-2 evolved recently by recombination in *Solanum* genus, which diverged from other solanaceous plants <6Mya[39].

## Discussion

We found that *Rcr3* is conserved across Solanaceae, whereas *Cf-2* belongs to a *Solanum*-specific clade of the *Hcr2* gene family (Fig. 8a). This indicates that *Cf-2* evolved by co-opting existing Rcr3 for Avr2 recognition. Interestingly, while *Rcr3* is present in Solanaceae, and most of these Rcr3 homologs can be inhibited by Avr2, only Rcr3 homologs from potato and tomato can effectively participate in *Cf-2*-dependent Avr2 recognition.

Most Rcr3 homologs from other plant species are sensitive to Avr2 inhibition. However, activity profiling experiments demonstrated that inhibition by Avr2 is strong for eggplant Rcr3, reduced for pepper and petunia Rcr3, and undetectable for tobacco Rcr3. Thus, Avr2 inhibition declines with the phylogenetic distance from tomato, consistent with the fact that Avr2 evolved in a tomato pathogen to inhibit tomato proteases. Furthermore, although eggplant, pepper and petunia Rcr3 are inhibited by Avr2, only *Solanum* Rcr3 can trigger HR when co-expressed with Cf-2. Thus, also Cf-2-dependent HR induction declines with the phylogenetic distance to tomato Rcr3, consistent with the evolution of Cf-2 in tomato by co-opting pre-existing Rcr3.

In contrast to all other tested Solanaceous plants, *N. benthamiana* lacks a functional Rcr3 protease. Interestingly, the repeated loss of *NbRcr3* in all *N. benthamiana* accessions and the loss of *NbPip1* in two accessions, suggests that these immune proteases were detrimental for *N. benthamiana*. We speculate

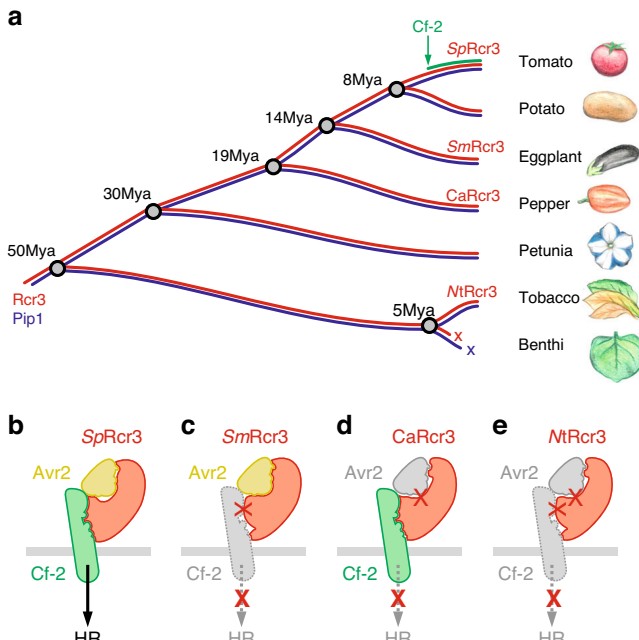

**Fig. 8 Model for evolution of Rcr3. a** Cf-2 evolved in wild tomato <8 Million years ago (<8Mya) by co-opting existing *Sp*Rcr3. Rcr3 and its paralog Pip1 evolved >50Mya and occur in all solanaceous species. *NbRcr3* and *NbPip1* in *N. benthamiana* carry deleterious mutations that occurred repeatedly in *N. benthamiana* species <5Mya. **b–e** Rcr3 needs to interact with both Avr2 and Cf-2 to trigger HR. **b** Both tomato and potato Rcr3 (e.g., *Sp*Rcr3) interact with Avr2 and Cf-2 and trigger HR. **c** Eggplant Rcr3 (*Sm*Rcr3) has a strong interaction with Avr2 but cannot trigger HR with Cf-2, presumably because it is unable to interact properly with Cf-2. **d** Pepper Rcr3 (*Ca*Rcr3) has a weak interaction with Avr2, but can trigger HR by introducing residues that enhance the interaction with Avr2. **e** Tobacco Rcr3 (*Nt*Rcr3) has low affinity to Avr2 and cannot trigger HR even when engineered to increase its interaction with Avr2.

that this is perhaps because *N. benthamiana* is an extremophile that adapted to the arid and harsh Australian climate. The production of immune proteases might have been a burden under conditions with a presumed low pathogen pressure. This is consistent with the notion that the *N. benthamiana* LAB accession also lacks several other immune-related genes[40], explaining its enhanced susceptibility to pathogens and its popularity for pathogen research.

Our data indicate that Rcr3 has two distinct interaction interfaces: with Avr2 and with Cf-2. Both these interactions are required to trigger HR (Fig. 8b). Both tomato and potato Rcr3 trigger HR and therefore interact well at both interfaces. Eggplant Rcr3 (*Sm*Rcr3) is unable to trigger HR despite its high affinity to Avr2, indicating that it poorly interacts with Cf-2 (Fig. 8c). By contrast, pepper Rcr3 (*Ca*Rcr3) can be engineered to trigger HR by introducing residues that enhance inhibition by Avr2,

suggesting that pepper Rcr3 is able to interact with Cf-2, and that only the interface with Avr2 is ineffective to trigger HR (Fig. 8d). However, a similar engineering approach for NbRcr3a-2 did not result in HR, indicating that in addition to low inhibition by Avr2, this protease is also unable to properly interact with Cf-2 (Fig. 8e). The inability to signal with Cf-2 most likely resides in a combination of species-specific residues. Nicotiana Rcr3, for instance, has 12 residues and one 3-amino acid deletion that are specific to Nicotiana (Fig. 4b, green).

There are various theories for Rcr3 interacting with Cf-2. First, only the Rcr3/Avr2 complex can bind to Cf-2. This simple model may or may not include a direct Avr2/Cf-2 interaction. Second, Rcr3 may interact with Cf-2 in a presignalling complex and Avr2 binding to Rcr3 would release the Avr2/Rcr3 complex so that Cf-2 would trigger HR. Third, Rcr3 remains bound to Cf-2 when binding Avr2 and a conformational change in Cf-2 would trigger HR. Unfortunately, we were unable to test these hypotheses and study the Rcr3/Cf-2 interaction, despite many attempts to establish interaction assays with purified proteins and using co-expression by agroinfiltration. However, when detecting Avr2-dependent HR we do have indirect evidence for a functional Rcr3/Cf-2 interaction.

We identified four residues in Rcr3 that facilitate Avr2 inhibition. Combining our data with previously published data on Avr2 inhibition of PLCPs[9,31,41], we can predict whether a PLCP is likely to be inhibited by Avr2 based on the following four rules: (i) a negatively charged residue at position 148; (ii) an uncharged small residue at position 151; (iii) an asparagine at position 194; and (iv) a positively charged residue at position 284. Indeed, we could show that introducing these residues into C14 and CYP3, which belong to PLCP subfamily-1 and -8, respectively, increases inhibition by Avr2. Strikingly, SpRcr3, which co-evolved with Cf-2 in S. pimpinellifolium, does not contain the residues for optimal inhibition by Avr2. There are several explanations for this counterintuitive observation.

First, this could be because Cf-2 only evolved recently, and therefore Rcr3 has not yet been able to adapt to its novel role as co-receptor. The occurrence of Cf-2 in only S. pimpinellifolium and the nearly identical nucleotide repeats within Cf-2[4] support that Cf-2 evolved only recently.

Second, the sensitivity of the Rcr3 co-receptor for Avr2 may already be sufficient to pass the threshold of recognition and trigger effective immunity with Cf-2, so no further adaptation for improved Avr2 inhibition was required. Still, to sense low Avr2 concentrations, the Avr2 binding affinity should be higher for Rcr3 than for the more abundant Pip1.

Third, Rcr3 might be a co-receptor to sense other pathogen-derived PLCP inhibitors and binding these inhibitors might require different residues. Indeed, nematode GrVap1 is also reported to trigger HR via Rcr3 and Cf-2[11]. Unfortunately, we were unable to establish GrVAP1-induced HR in our assays hence we could not investigate dual selection on residues promoting Avr2 inhibition.

A fourth, more compelling hypothesis is that Rcr3 is under dual selection to prevent Cf-2-dependent auto-immune responses. For instance, R284 in the auto-necrosis-inducing allele SlRcr3 promotes Avr2 inhibition, whereas S. pimpinellifolium Rcr3 (SpRcr3), which co-evolved with Cf-2, carries Q284. We therefore speculate that Q284 may prevent the interaction with a host-derived protein, and thereby prevent Cf-2-dependent auto-necrosis. Thus, the occurrence of Q284 in Rcr3 only in S. pimpinellifolium[32], may be the result of selection against auto-necrosis in tomato species carrying Cf-2.

A final hypothesis is that residues in Rcr3 that weaken Avr2 inhibition are there to reduce interactions with other pathogen-derived inhibitors, allowing Rcr3 to act in basal immunity in the absence of Cf-2. This hypothesis is supported by the fact that polymorphic sites in tomato Rcr3 are also polymorphic in solanaceous plants that do not have Cf-2 and are not susceptible to C. fulvum producing Avr2. Indeed, in the absence of Cf-2, Rcr3 does contribute to basal immunity against P. infestans[8,10] and P. infestans secretes EPIC inhibitors that inhibit Rcr3 but escape recognition via Cf-2[8]. For instance, D148 in SlRcr3 promotes inhibition by both Avr2 and PmEPIC1 of P. mirabilis[33]. The H148D substitution in other Rcr3 alleles might suppress inhibition by PmEPIC1 and similar pathogen-derived inhibitions.

Our data support the hypothesis that Cf-2 recently evolved in wild tomato. Cf-2 is a member of a clade-1 Hcr2 subfamily that only exists in Solanum (potato and tomato). Cf-5 is another member of this Solanum-specific clade-1 Hcr2s. We demonstrated that Cf-5-dependent recognition of Avr5 does not require Rcr3, or the closely related PLCPs Pip1 and Pfp1 because co-expression of Cf-5 and Avr5 triggers HR in N. benthamiana, which lacks Rcr3, Pip1 and Pfp1. Since the specificity-determining component of Cf-5 and Cf-2 must reside in the variable LRR region we conclude that the LRR region interacts with Rcr3, and that Cf-2 is unique within the Hcr2 family in its requirement for Rcr3. Therefore, we propose that when Cf-2 evolved, pre-existing Rcr3 gained a novel molecular function as a co-receptor for the recognition of Avr2, in addition to its role in basal immunity. The recent evolution of Cf-2 is also supported by the fact that it contains a tandem repeat of nearly identical nucleotide sequence causing recombination instability[4], a feature that would be counter selected.

The evolution mechanism of an immune receptor co-opting an existing effector-target contrasts with that of ZAR1 and Prf: two highly conserved NLRs in eudicots and solanaceae, respectively. Arabidopsis ZAR1 is a NLR that interacts with decoys of receptor-like cytoplasmic kinases (RLCKs) ZED1, ZRK3, and RKS1 for the recognition of the P. syringae effectors HopZ1a and HopF2a, and the Xanthomonas campestris effector AvrAc, respectively[42–45]. The N. benthamiana ZAR1 homolog interacts with the ZED1-like homolog JIM2 for the recognition of the X. perforans effector XopJ4[46]. Thus, the ZAR1 receptor is conserved across species, but the guarded decoy (pseudo)kinases ZED1, RKS1, ZRK3, and JIM2 appear to be lineage-specific, in contrast to Cf-2 and Rcr3 where the receptor evolved recently and the decoy is conserved. Similarly, the tomato immune receptor Prf recognizes the interaction of the P. syringae effectors AvrPto and AvrPtoB with kinase Pto[47]. Also here, Prf is conserved across Solanaceae[19] while Pto and its homologs are evolving.

As with Cf-2/Rcr3, polymorphisms in the guarded decoy is also important for ZAR1/ZED1 and Prf/Pto interactions. For example, co-expression of ZED1 with HopZ1a by agroinfiltration is sufficient to trigger HR in N. benthamiana, and this HR depends on the endogenous N. benthamiana ZAR1[48]. However, co-expression of Arabidopsis ZAR1 with XopJ4 in a N. benthamiana zar1 knockout line is not sufficient to trigger HR[46], indicating that N. benthamiana ZED1-like homolog is unable to interact with AtZAR1.

Likewise, transgenic N. benthamiana carrying 35S-driven Pto are resistant to P. syringae pv. tabaci strains expressing AvrPto[49] and HR in these lines depend on the endogenous N. benthamiana Prf homolog[50]. However, this transgenic line does not confer resistance to P. syringae expressing AvrPtoB[51], indicating that Pto does not interact with NbPrf for the recognition of AvrPtoB. Furthermore, transgenic N. benthamiana expressing Pto from its native promoter are not resistant to P. syringae strains carrying either AvrPto or AvrPtoB, unless these lines also express tomato Prf[51]. These data indicate that Pto has evolved to interact with tomato Prf and that only Pto overexpression can force the interaction with NbPrf to recognise AvrPto, but not AvrPtoB.

In conclusion, our study highlights how a new receptor can give an ancient protein a new function as co-receptor. The Cf-2 receptor has evolved very recently, unlike ZAR1 and Prf. We speculate that indirect recognition mechanisms initially evolve in a manner similar to *Cf-2*. Recognition spectra are subsequently expanded by the evolution of additional decoys, which retain a degree of specificity for the available receptor homolog they evolve with. Understanding the evolution of defence components is crucial for the future rational engineering and application of these genes.

## Methods

**Cloning of general binary cloning vectors**. The Golden Gate Modular Cloning (MoClo) kit[52] and the MoClo plant parts kit[53] were used for cloning, and all vectors are from this kit unless specified otherwise. Cloning design and sequence analysis were done using Geneious Prime 2020.0.4 (https://www.geneious.com). Plasmid construction is described in Supplementary Data 3.

**Sequence retrieval**. Tomato *Rcr3* (Solyc02g076980.2), *Pip1* (Solyc02g077040.2), *P1L* (Solyc03g006210.1), *Pfp1* (Solyc02g077050.2), *Cf-9* (Genbank LPU15936), and the 3′ conserved region of *Cf-2* (Genbank U42445.1) were used to retrieve homologous genes in other organisms by BLAST searches in Solgenomics against the available genomes and databases. Additional sequences were obtained by BLAST searches against the NCBI RefSeq proteomes. Sequences, which had not been annotated or were incorrectly annotated, were manually annotated. All sequences are described in Supplementary Data 2. *N. benthamiana* PLCP genes were manually reannotated and compared to those found in the recently reannotated *N. benthamiana* gene models[27].

**Manual reannotation of benthi PLCPs**. Amino acid sequences were extracted from the NbDE and Niben1.0.1 annotations based on the Pfam peptidase C1 domain (PF00112). Exhaustive tblastn using the identified PLCPs as a query and the Niben1.0.1 genome as subject was used to identify partial/not annotated PLCPs. Gene models were compared to the top five highest similarity genes identified by blastn with the NCBI RefSeq *Nicotiana* species as subject. Where needed, exon–exon boundaries and transcription start and termination sites of gene models were adjusted based on the NCBI RefSeq annotation. Truncated *PLCP* genes with frame-shift mutations were cross-validated against the independent Niben0.4.4, Nbv0.5, and Nbv0.3 genome assemblies to ensure that the mutation was not an artefact of genome assembly.

**Tree building**. Clustal Omega[54,55] was used to align sequences. Determining the best model for maximum likelihood, phylogenetic analysis was performed in MEGA X as indicated in the text[56] or using FastTreeMP[57].

**E. coli expression vector cloning and Avr2 purification**. For protein expression pJK153 (pET28b-T7::OmpA-HIS-TEV-Avr2) and pSM101 (pET28b-T7::OmpA-HIS-TEV-Avr2Δ6) were transformed to *E. coli* strain Rossetta2(DE3)pLysS (Novagen/Merck). An overnight grown starter culture was diluted 1/100 in TB [per liter: 24 g yeast extract, 12 g Bacto-tryptone, 0.5% glycerol, after autoclaving add 0.2 M/L potassium phosphate buffer [26.8 g $KH_2PO_4$ and 173.2 g $K_2HPO_4$ pH 7.6], grown for ~3 h to $OD_{600} \approx 0.6$ and induced overnight at 37 °C by the addition of 0.8 mM isopropyl β-D-1-thiogalactopyranoside (IPTG). Afterwards the supernatant was collected by centrifugation of the bacterial culture at $10,000 \times g$, 4 °C for 20 min. The supernatant was adjusted to 50 mM Tris-HCl, 150 mM NaCl, 10 mM imidazole and run over a Ni-NTA agarose (Qiagen)-loaded column by gravity-flow. The Ni-NTA agarose was washed with 10 column volumes of washing buffer (50 mM Tris-HCl, 150 mM NaCl, 50 mM imidazole, pH 8) and the purified protein was eluted in elution buffer (50 mM Tris-HCl, 150 mM NaCl, 250 mM imidazol, pH 8). Finally, the protein was concentrated by centrifugation in 3 kDa cut-off filters (Amicon Ultra-15 Centrifugal Filter Units, Merck) at $4500 \times g$, 4 °C, flash-frozen in liquid nitrogen and stored at −80 °C. Protein concentration was determined by Bradford assay using BSA as a standard, and purity was verified by running the protein on 18% SDS-PAGE gels followed by Coomassie staining[58].

**Agroinfiltration**. For transient expression of proteins in *N. benthamiana*, *A. tumefaciens* strain GV3101 pMP90 carrying binary vectors were inoculated from glycerol stock in LB supplemented with 25 μg/ml rifampicin, 50 μg/ml gentamycin and 50 μg/ml kanamycin and grown O/N at 28 °C until saturation. Cells were harvested by centrifugation at $2000 \times g$, RT for 5 min. Cells were resuspended in infiltration buffer (10 mM $MgCl_2$, 10 mM MES-KOH pH 5.6, 200 μM acetosyringone) to an $OD_{600} = 0.25$ unless stated otherwise and left to incubate in the dark for 2 h at RT prior to infiltration into 5-week-old *N. benthamiana* leaves.

**HR assays**. For HR assays, 5-week-old *N. benthamiana* plants were agroinfiltrated with the stated combinations of binary vector containing Agrobacterium at $OD_{600} = 0.25$ each. Images were taken at 5 dpi unless stated otherwise.

**ABPP**. Apoplastic fluid (AF) was isolated from agroinfiltrated *N. benthamiana* plants expressing different proteins at 2–3 days post-infiltration[26]. Briefly, the AF was extracted by vacuum infiltrating *N. benthamiana* leaves with ice-cold MilliQ. Leaves were dried with filter paper to remove excess liquid, and apoplastic fluid was extracted by centrifugation of the leaves in a 20 ml syringe barrel (without needle or plunger) in a 50 ml falcon tube at $2000 \times g$, 4 °C for 25 min. Samples were used directly in labelling reactions. Samples were adjusted to 5 mM DTT, 50 mM NaAc pH 5.0, and preinhibited for 45 min with either 100 μM E-64 (a general PLCP inhibitor), 2–4 μM purified Avr2 or Avr2Δ6, or 1% DMSO (final concentration) as a control. After preinhibition, samples were labelled with 2 μM MV201[22] for 3 h at room temperature (total volume 50 μl). The labelling reaction was stopped by precipitation with 5 volumes of ice-cold acetone, followed by a 10 s vortex and immediate centrifugation at $16,100 \times g$, 4 °C for 5 min. Samples were resuspended in 2× sample buffer [100 mM Tris-HCl (pH 6.8), 200 mM DTT, 4% SDS, 0.02% bromophenol blue, 20% glycerol] and boiled for 5 min 95 °C prior to running on 15% SDS-PAGE gels. Fluorescence scanning was performed on an Amersham Typhoon scanner (GE Healthcare), using Cy3 settings. Equal loading was verified by Coomassie staining[58].

**CRISPR/Cas9 construct design**. Two CRISPR/Cas9 sgRNAs targeting *NbPip1* (Niben101Scf08921g02024.1) at two positions 90 bp apart were designed to minimize in silico off-targets[59] with maximum in silico on-target activity[60]. Specific PCR products were generated by amplifying from the template for sgRNAs pICH86966[25] (Addgene #46966) using forward primers 5′-tgtggtctcaattgtgactgcatgattcagcgtgtttagagctagaaatagcaag-3′ (for sgRNA_25) or 5′-tgtggtctcaattgaaaattgttagagatgccggtttagagctagaaatagcaag-3′ (for sgRNA_44) and reverse primer 5′-tgtggtctcaagcgtaatgccaactttgtac-3′. The two PCR products were combined with the AtU6 promoter (pICSL01009[25], Addgene #46968), and pICH47751 or pICH47761 (Addgene #48002 and #48003) in a Golden Gate reaction with BsaI to generate pJK084 (pL1M-F3-AtU6p::sgRNA>*Nb*Pip1_25) and pJK085 (pL1M-F3-AtU6p::sgRNA>NbPip1_44) respectively. The final binary CRISPR/Cas9 vector (pL2M-KAN-Cas9 > *Nb*Pip1) was constructed by combining pAGM4723 and pICH41780 (Addgene #48015 and #48019) with pICSL11017 (Addgene #51144), pICH47742[61] (Addgene #49771), and pJK084 and pJK085 in a Golden Gate reaction with BpiI, resulting in pJK090. Sequences of sgRNA_25 (*Nb*Pip1 position 1) 5′-gtgactgcatgattcagcgtgtttagagctagaaatagcaagttaaatataaggctagtccgttatcaacttgaaaaagtggcaccgagtcggtgctttttt-3′ and sgRNA_44 (*Nb*Pip1 position 2) 5′-gaaaattgttagagatgccggtttagagctagaaatagcaagttaaaataaggctagtccgttatcaacttgaaaaagtggcaccgagtcggtgctttttt-3′ (underlined: spacer sequence).

**Nicotiana benthamiana tissue culture**. Agrobacterium-mediated transformation of *N. benthamiana* was performed as described[62]. Briefly, 4-week-old *N. benthamiana* plants were agroinfiltrated with *A. tumefaciens* strain GV3101 pMP90 carrying binary plasmid pJK090 (pL2M-KAN-Cas9 > *Nb*Pip1) at $OD_{600} = 0.2$. Three days post-infiltration, infiltrated leaves were cut into 1 cm² sections and cultured under sterile conditions in shooting media [per liter: 4.3 g MS basal salts (1×), 30 g sucrose, 1× B5 vitamins, 0.1 mg Naphthalene Acetic Acid (NAA), 0.59 g MES, pH = 5.7 using KOH, 4 g Agargel, after autoclaving add 1 mg Benzylaminopurine (BAP), 300 mg Ticarcillin/clavulanic acid] for 3 days, and subsequently transferred to shooting media supplemented with 150 μg/ml kanamycin to select for stable transformants and were further cultured until shoots appeared. Shoots >3 mm were excised and transferred to rooting media [per liter: 2.15 g MS basal salts (0.5×), 5 g sucrose, 0.1 mg NAA, pH = 5.8 with KOH, 2.5 g Gelrite, after autoclaving add 300 mg Ticarcillin/clavulanic acid] supplemented with 150 μg/ml kanamycin and were further cultured until roots appeared, after which plantlets were transferred to soil and grown in the growth chamber. T0 *N. benthamiana* transformants were screened for *NbPip1* mutant alleles by sanger sequencing of PCR fragments generated using forward primer: 5′-actaatctcactttcggagcaag-3′ and reverse primer 5′-ccacacattcttgtcgaacaacct-3′. *NbPip1* mutant T1 and T2 lines lacking the T-DNA containing the kanamycin resistance and *Cas9* transgenes were confirmed by PCR on the *NPTII* using forward primer: 5′-gaggctattcggctatgactgg-3′ and reverse primer 5′-tatgtcctgatagcggtccg-3′ and these lines were used for further experiments.

**VIGS**. A partially domesticated TRV2 vector compatible with Golden Gate cloning using BsaI (pJK037) was generated by combining the PCR fragments derived from the plasmid template pTRV2[63]. using the forward 5′-ttggtctcaggatccggtaccgagctcacgc-3′ and reverse 5′-ttggtctcaaggccttttcgacctttttcccctgc-3′ and forward 5′-ttggtctcagcctcttttcctgtggatagcacgtac-3′ and reverse 5′-ttggtctcagaattcggtaaccttactcacagaatc-3′ primers with the PCR fragment derived from the plasmid template pFK210B-AtMIR390a-B/c[64] (Addgene #51777) using the forward 5′-aaaaagaattcagagaccattaggcaccccagg-3′ and reverse 5′-aaaaaggatcctgagaccgtcgaggtgcagactgg-3′ in a BsaI Golden Gate reaction. Two independent VIGS constructs targeting *NbPip1* were ordered as GeneArt GeneStrings (ThermoFisher) and combined with pJK037 in a BsaI Golden Gate reaction generating pJK220 (pL2M-

**13**

TRV2 > *Nb*Pip1#1) containing a 300 bp fragment targeting *NbPip1* and pJK221 (pL2M-TRV2 > *Nb*Pip1#2) containing a previously published 576 bp fragment targeting *NbPip1*[24]. *A. tumefaciens* strain GV3101 pMP90 carrying binary plasmid pJK220 (pL2M-TRV2::*Nb*Pip1#1), pJK221 (pL2M-TRV2::*Nb*Pip1#2), or *TRV2::GFP* were mixed 1:1 with bacteria containing binary plasmid TRV1[55] at a combined final OD$_{600}$ = 1 in infiltration buffer as described above. The mixed cultures were infiltrated into leaves of 10-day-old *N. benthamiana* plants. The infiltrated seedlings were grown for 3–4 weeks in a growth chamber until use.

**Colony count.** *Pseudomonas syringae* pv. *tomato* DC3000 ΔhopQ1-1[65] was grown to saturation from glycerol stock O/N at 28 °C in LB supplemented with 25 µg/ml rifampicin, washed and resuspended in 10 mM MgCl$_2$ at OD$_{600}$ = 0.0002 ($1 \times 10^5$ CFU/ml) and were used for infection of plants by syringe infiltration. For CRISPR/Cas9 *NbPip1* KO lines, 4-week-old mutant or wild-type plants were infected, whereas for VIGS silencing of *NbPip1* plants were infected 3 weeks post infection with either pJK220 (pL2M-TRV2::*Nb*Pip1#1), pJK221 (pL2M-TRV2::*Nb*Pip1#2), or *TRV2::GFP*. Three leaf discs (8 mm diameter) were punched from the infected leaves of four independent plants for each treatment at 3 h, 24 h, 48 h, and 72 h post infection and were incubated in 15% H$_2$O$_2$ for 5 min to sterilise leaf surfaces. Leaf discs were washed twice in sterile MilliQ and ground in 1 ml sterile MilliQ for 5 min by addition of two metal beads in a tissuelyser (Biospec Products, Bartlesville, OK, USA). Serial dilutions of the homogenate were plated onto LB agar plates containing 25 µg/ml rifampicin. Colonies were counted after 48 h incubation at 28 °C. Statistics were performed in R.

**Reporting summary.** Further information on research design is available in the Nature Research Reporting Summary linked to this article.

## Data availability

All data supporting the findings of this study are available in the manuscript and its supplementary files. The data underlying Figs. 2c, 3f, 5a, 6a–d and Supplementary Figs. 5 and 6a, b are provided as a Source File. All alignments (fasta files) and trees (newick files) have been deposited on Oxford Research Archive at https://doi.org/10.5287/bodleian: VJwe6gBNx. Phylogenetics trees of Figs. 3g, 4a, 7, S2, S4, S7, S8 can also be viewed on iTOL (https://itol.embl.de/shared/JKourelis). Source data are provided with this paper.

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

## Acknowledgements

We thank Matthieu Joosten for sending the Cf-2::EGFP and Cf-4::EGFP plasmids; the ENSA consortium for providing general Golden Gate-compatible plasmids; Ursula Pyzio for excellent plant care; John Baker for photography; Jonathan Jones, Sophien Kamoun, James Carrington, Sylvestre Marillonnet, and Nicola Patron for providing plasmids via AddGene. This work has been supported by 'The Clarendon Fund' (JK), the Oxford Indira Gandhi Graduate Scholarship (SM), ERC Consolidator grant 616449 'Green-Proteases' (RvdH, JK), and BBSRC grant BB/S003193/1 'Pip1S' (RH). The funders had no role in study design, data collection and analysis, decision to publish, or preparation of the manuscript.

## Author contributions

Conceptualization: J.K., J.P., and R.v.d.H.; Formal analysis: J.K.; Investigation: J.K., S.M.; O.M., S.K., and P.S.K. under supervision of J.K.; Funding acquisition: R.v.d.H.; Writing initial draft: J.K.; Editing: J.K., R.v.d.H.

## Competing interests

The authors declare no competing interests.
