## [Peer Review File · Nature Communications]

REVIEWER COMMENTS

Reviewer #1 (Remarks to the Author):

This manuscript discusses the evolution of a guarded decoy – RLP receptor pair (Rcr3 and Cf-2) in Solanaceae plants.

First the authors verified the system to test whether effector-decoy-receptor combination is functional with cell death as the indicator after expressing the effector Avr2, the guarded decoy Rcr3, and the receptor Cf-2 by agroinfiltration in *Nicotiana benthamiana* (Fig1). Then, they showed that *N. benthamiana* Rcr3 has frameshift mutations causing no functional protein accumulation. Even if the frameshifts were corrected, *N. benthamiana* Rcr3 is shown to be non-functional for Cf-2-mediated cell death (Fig2). In addition, Rcr3 homolog, Pip1, has also mutations in certain *N. benthamiana* accessions including the LAB accession (Fig3). Then, they tested functions of Rcr3 from various Solanaceae species for suppression of Rcr3 activity by Avr2 and for Cf-2-mediated cell death (Fig4 and 5). Finally, they compared Hcr2-related gene sequences including Cf-2 and Cf-5 from various Solanaceae species (Fig6).

The authors performed each experiment and analysis very nicely and the manuscript is carefully written. Each result can be potentially interesting. However, experiments and analyses in this manuscript were not integrated well. For example, most of their conclusions regarding evolution do not need their experiments. They investigated important amino acid residues in Rcr3 which define functionality for inhibition by Avr2 and for Cf-2 mediated cell death, but those experiments, unfortunately, did not help understand the evolution of Rcr3 or Avr2-Rcr3-Cf-2. For example, some of Rcr3 versions that were inhibited by Avr2 could not trigger Cf-2-mediated cell death. Thus, it remains unclear what are critical evolution in Rcr3 for compatibility with Cf-2 mediated immunity. Also, what is the physiological significance of Rcr3 sequence variation, then?

Recent and independent loss of Rcr3 in *N. benthamiana* can be interesting but the authors did not address evolutionary pressure that caused this or evolutionary consequence of Rcr3 loss. Also, loss of Pip1 can be interesting but it is rather descriptive at this stage.

The authors used Rcr3 inhibition by Avr2 and Rcr3 binding/interaction interchangeably and without showing data, which is problematic.

In summary, the research topic is very important and interesting and experiments were carefully performed but experiments and analyses in this manuscript were not integrated well to make strong conclusions.

Reviewer #2 (Remarks to the Author):

This paper provides a detailed evolutionary and functional analysis of the Avr2/Cf2/Rcr3 system. The authors establish a robust assay for Cf2-mediated activation of hypersensitive response in *Nicotiana benthamiana* / *Nicotiana tabacum* and investigate several native variants of Rcr3. At the end they pinpoint Avr2 inhibition specificity to four residues. Finally, they investigate evolution of Cf2 across Solanaceae and conclude that it evolved recently, after the decoy was already present and the clade it belongs to mostly diversifies through repeat copy number variation.

Overall, this is a high quality evolutionary and functional analysis of an important existing system. Understanding the presented system can lead to new engineering approaches that would be of broad interest in plant pathology.

As we are still in the midst of the COVID19 situation, I would make most of my comments about the data presented here rather than asking for additional experiments which might or may not be needed depending on the answer to my first point below.

I have several specific questions that need to be clarified:

1. Correct identification of all orthologs and paralogs can affect some of the conclusions of this paper. *Nicotiana benthamiana* is a natural allotetraploid (as mentioned on line 190). This should be brought up earlier and there should be a clear description of homeologous copies of Rcr3 and Pip1. Are they 100% identical? If so this needs to be mentioned. If not, investigating only one of the copies might not be enough to draw functional or evolutionary conclusions. As *Nicotiana benthamiana* genome is incomplete, specific caution should be used and additional resources should be deployed (such as 1KP project <https://sites.google.com/a/ualberta.ca/onekp/>) to look at the evolution of these genes.
2. Although I think I understand the logic of how authors arrived at 4 sites they manipulate to investigate differential inhibition by Avr2, I would like to see all differences between orthologs to be discussed first, including showing a full aa alignment and its schematic representation in the figure 2b,c and then explaining how these 4 residues were deduced. I might be missing it, but I think these are not the only variations across the sequences and there could be other sites that are important.
3. I really like the visual representation of critical residues on a structural model in Figure 5, however I would also like to see this model colored by general conservation across the alignment of all sequences in Figure 4 (see my point above). Do to this, you can use following guidelines: <https://www.cgl.ucsf.edu/chimera/data/tutorials/systems/outline.html>
4. A minor comment is about the nomenclature of alleles. Sometimes a prefix is used as in NbRcr3 and sometimes a superscript. The superscript is hard to read, so I would suggest converting everything to prefix even though there might be historical reasons for double convention.
5. In materials and methods, there is mention of manual re-annotation of genes (lines 453-454). Please, include more details how this was done.
6. Please, include all multiple sequence alignments used in the paper (also those to make the trees) as supplemental datasets or make them available on Figshare.
7. Please, make all tree files available, ideally in interactive mode on iTOL.

With very best wishes to all authors!

Ksenia Krasileva

We like to thank the reviewers for their constructive criticism. We have used their remarks to improve the manuscript.

Reviewer #1 (Remarks to the Author):

This manuscript discusses the evolution of a guarded decoy – RLP receptor pair (Rcr3 and Cf-2) in Solanaceae plants.

First the authors verified the system to test whether effector-decoy-receptor combination is functional with cell death as the indicator after expressing the effector Avr2, the guarded decoy Rcr3, and the receptor Cf-2 by agroinfiltration in *Nicotiana benthamiana* (Fig1). Then, they showed that *N. benthamiana* Rcr3 has frameshift mutations causing no functional protein accumulation. Even if the frameshifts were corrected, *N. benthamiana* Rcr3 is shown to be non-functional for Cf-2-mediated cell death (Fig2). In addition, Rcr3 homolog, Pip1, has also mutations in certain *N. benthamiana* accessions including the LAB accession (Fig3). Then, they tested functions of Rcr3 from various Solanaceae species for suppression of Rcr3 activity by Avr2 and for Cf-2-mediated cell death (Fig4 and 5). Finally, they compared Hcr2-related gene sequences including Cf-2 and Cf-5 from various Solanaceae species (Fig6).

The authors performed each experiment and analysis very nicely and the manuscript is carefully written. Each result can be potentially interesting. However, experiments and analyses in this manuscript were not integrated well. For example, most of their conclusions regarding evolution do not need their experiments. They investigated important amino acid residues in Rcr3 which define functionality for inhibition by Avr2 and for Cf-2 mediated cell death, but those experiments, unfortunately, did not help understand the evolution of Rcr3 or Avr2-Rcr3-Cf-2.

RESPONSE: We have revised the text at multiple places to integrate this analysis more from an evolutionary angle. The experiments describing that only one of the four residues promoting Avr2 inhibition are present in the Rcr3 allele that evolved with Cf-2 is very interesting from an evolutionary standpoint because this is counterintuitive. We find it essential to explain this enigma accurately as they imply that other evolutionary forces are at play on Rcr3, as we now indicate in the introduction and hypothesise in the discussion. This observation is relevant also for other indirect perception mechanisms.

For example, some of Rcr3 versions that were inhibited by Avr2 could not trigger Cf-2-mediated cell death. Thus, it remains unclear what are critical evolution in Rcr3 for compatibility with Cf-2 mediated immunity.

RESPONSE: We thank the reviewer for this helpful suggestion. We have summarised the polymorphic residues in the protease domain of Rcr3 from different species in **NEW Figure 4b**. We have highlighted residues that facilitate Avr2 inhibition. We have also highlighted residues in *Nicotiana* Rcr3 that collectively prohibit HR signalling (green in **NEW Figure 4b**). These residues are scattered over the structural model and identifying which of these residues are important for HR signalling is the subject for future studies.

Also, what is the physiological significance of Rcr3 sequence variation, then?

RESPONSE: The physiological significance of Rcr3 sequence variation is now better explained in the introduction and is multifaceted as it may affect interactions with substrates, inhibitors and Cf-2. Our data demonstrates that variation in Rcr3 separates

the interactions with Avr2 and Cf-2 because we have Rcr3 variants that do not trigger Cf-2-dependent HR, even though they interact with Avr2. We discuss the putative roles of the observed variation in tomato Rcr3 again when we give explanations for the fact that *SpRcr3* does not have the residues required for maximal Avr2 inhibition.

Recent and independent loss of Rcr3 in *N. benthamiana* can be interesting but the authors did not address evolutionary pressure that caused this or evolutionary consequence of Rcr3 loss. Also, loss of Pip1 can be interesting but it is rather descriptive at this stage.

RESPONSE: We have added a paragraph on the loss of Rcr3 and Pip1 in *N. benthamiana* in the discussion section. We speculate that this is because *N. benthamiana* is an extremophile that has specialised to arid conditions on the Australian continent. We speculate that disease pressure might be low and the production of immune proteases might be a burden under these conditions.

The authors used Rcr3 inhibition by Avr2 and Rcr3 binding/interaction interchangeably and without showing data, which is problematic.

RESPONSE: We agree with the reviewer that it is more accurate to make claims on inhibition of Rcr3 labelling, which is our main readout. We have adjusted the statements where appropriate.

In summary, the research topic is very important and interesting and experiments were carefully performed but experiments and analyses in this manuscript were not integrated well to make strong conclusions.

RESPONSE: We thank the reviewer for these comments. We trust that the made revisions have improved the integration of the experiments and analyses.

Reviewer #2 (Remarks to the Author):

This paper provides a detailed evolutionary and functional analysis of the Avr2/Cf2/Rcr3 system. The authors establish a robust assay for Cf2-mediated activation of hypersensitive response in *Nicotiana benthamiana* / *Nicotiana tabacum* and investigate several native variants of Rcr3. At the end they pinpoint Avr2 inhibition specificity to four residues. Finally, they investigate evolution of Cf2 across Solanaceae and conclude that it evolved recently, after the decoy was already present and the clade it belongs to mostly diversifies through repeat copy number variation.

Overall, this is a high quality evolutionary and functional analysis of an important existing system. Understanding the presented system can lead to new engineering approaches that would be of broad interest in plant pathology.

As we are still in the midst of the COVID19 situation, I would make most of my comments about the data presented here rather than asking for additional experiments which might or may not be needed depending on the answer to my first point below.

RESPONSE: We appreciate the enthusiastic support of this reviewer and for not asking for additional experiments given the situation. The COVID19 situation in the UK is still dire and most labs are still closed in Oxford.

I have several specific questions that need to be clarified:

1. Correct identification of all orthologs and paralogs can affect some of the conclusions of this paper. *Nicotiana benthamiana* is a natural allotetraploid (as mentioned on line 190). This should be brought up earlier and there should be a clear description of homeologous copies of Rcr3 and Pip1. Are they 100% identical? If so this needs to be mentioned. If not, investigating only one of the copies might not be enough to draw functional or evolutionary conclusions. As *Nicotiana benthamiana* genome is incomplete, specific caution should be used and additional resources should be deployed (such as 1KP project <https://sites.google.com/a/ualberta.ca/onekp/>) to look at the evolution of these genes.

RESPONSE: We thank the reviewer for this suggestion. We now discuss the allopolyploidy of *N. benthamiana* earlier and describe the *Rcr3* and *Pip1* genes in this species more accurately. We identified two *NbRcr3*'s and one *NbPip1* in *N. benthamiana*. These three genes were detected in two independent genome sequences (Niben101 and NbV05) and are also present in the new, unpublished genome assembly (nbenth.com) which is based on long-read sequencing. We see these three genes also in the genome of the QLD accession, and we were able to clone them from the LAB accession. These three genes contain frameshift mutations in all analysed genome assemblies and in the cloned sequences. We have added the nt and aa alignments of these three genes with the tomato orthologs to illustrate their high conservation (**NEW Supplemental Figure S1**). Upon ORF correction, they phylogenetically cluster with Rcr3 and Pip1, respectively (**Supplemental Figure S2**), leaving no doubt that these are orthologs of Rcr3 and Pip1, respectively.

2. Although I think I understand the logic of how authors arrived at 4 sites they manipulate to investigate differential inhibition by Avr2, I would like to see all differences between orthologs to be discussed first, including showing a full aa alignment and its schematic representation in the figure 2b,c and then explaining how these 4 residues were deduced. I might be missing it, but I think these are not the only variations across the sequences and there could be other sites that are important.

RESPONSE: The reviewer is right to ask for these alignments as they are very helpful to explain our reasoning. We have now added a summary of the polymorphic residues in the protease domain as **NEW Figure 4b**.

3. I really like the visual representation of critical residues on a structural model in Figure 5, however I would also like to see this model colored by general conservation across the alignment of all sequences in Figure 4 (see my point above). Do to this, you can use following

guidelines: www.cgl.ucsf.edu/chimera/data/tutorials/systems/outline.html

RESPONSE: We thank the reviewer for this useful suggestion. We have added a structural model of Rcr3 with conservation highlighted using ConSurf (Ashkenazy et al., NAR 2016) as **NEW Figure 4c**.

4. A minor comment is about the nomenclature of alleles. Sometimes a prefix is used as in NbRcr3 and sometimes a superscript. The superscript is hard to read, so I would suggest converting everything to prefix even though there might be historical reasons for double convention.

RESPONSE: We agree that the consistency of the nomenclature could be improved and that the superscript is hard to read. We have therefore revised the text and figures to

remove the superscript and use prefix consistently for species abbreviations and substitution mutations between brackets.

5. In materials and methods, there is mention of manual re-annotation of genes (lines 453-454). Please, include more details how this was done.

RESPONSE: We have now added a paragraph to the Methods describing the manual re-annotation of *N. benthamiana* *PLCP* genes, as requested.

6. Please, include all multiple sequence alignments used in the paper (also those to make the trees) as supplemental datasets or make them available on Figshare.

RESPONSE: We have added the sequence alignments as supplemental files in fasta format.

7. Please, make all tree files available, ideally in interactive mode on iTOL.

RESPONSE: We have uploaded the tree files in iTOL and added them as supplemental files in newick format.

REVIEWERS' COMMENTS:

Reviewer #1 (Remarks to the Author):

The revised manuscript has been greatly improved in terms of integration of their experiments into evolutionary scenario. I have two further suggestions.

First, the authors still use "interaction" without testing interaction in some places such as in Figure 6, so they should carefully describe to avoid confusion by readers. In addition, "This indicates that HR signalling can be re-instated in CaRcr3 by enhancing the Rcr3-Avr2 interaction." (L343-344). Rcr3 inhibition by Avr2 would need an interaction but strength of inhibition by Avr2 does not necessarily indicate strength of interaction. I think this is an important point.

Second, in the case of Pepper Rcr3, Rcr3 inhibition by Avr2 and Cf-2-mediated HR are correlated (L335-347). However, in other cases, Rcr3 inhibition by Avr2 is clearly NOT sufficient for Cf-2-mediated HR (Eggplant Rcr3 and engineered *N. benthamiana* Rcr3). Therefore, I wonder if the authors would like to discuss "Second, the sensitivity of the Rcr3 co-receptor for Avr2 may already be sufficient to pass the threshold of recognition and trigger effective immunity with Cf-2, so no further adaptation for improved Avr2 inhibition was required." I think that the authors simply don't know at this stage what (inhibition and/or interaction and/or something else) is important for Cf-2-mediated HR. Hopefully, the authors can develop experimental systems to test Rcr3-Cf-2 interactions to fully address this issue in the future.

I hope that the COVID-19 situation there will be better soon.

Reviewer #2 (Remarks to the Author):

The authors addressed all of my comments. Well done!

REVIEWERS' COMMENTS:

Reviewer #1 (Remarks to the Author):

The revised manuscript has been greatly improved in terms of integration of their experiments into evolutionary scenario. I have two further suggestions. **RESPONSE:** Thanks for acknowledging that the evolutionary scenario has been improved greatly.

First, the authors still use “interaction” without testing interaction in some places such as in Figure 6, so they should carefully describe to avoid confusion by readers. In addition, “This indicates that HR signalling can be re-instated in CaRcr3 by enhancing the Rcr3-Avr2 interaction.” (L343-344). Rcr3 inhibition by Avr2 would need an interaction but strength of inhibition by Avr2 does not necessarily indicate strength of interaction. I think this is an important point. **RESPONSE:** We have carefully read this section again and made several revisions to be most accurate. We do not fully understand how the strength of inhibition may not indicate the strength of interaction. Please note that we used ‘indicates’ in the cited sentence, which leaves sufficient wiggle room.

Second, in the case of Pepper Rcr3, Rcr3 inhibition by Avr2 and Cf-2-mediated HR are correlated (L335-347). However, in other cases, Rcr3 inhibition by Avr2 is clearly NOT sufficient for Cf-2-mediated HR (Eggplant Rcr3 and engineered *N. benthamiana* Rcr3). Therefore, I wonder if the authors would like to discuss “Second, the sensitivity of the Rcr3 co-receptor for Avr2 may already be sufficient to pass the threshold of recognition and trigger effective immunity with Cf-2, so no further adaptation for improved Avr2 inhibition was required.” I think that the authors simply don’t know at this stage what (inhibition and/or interaction and/or something else) is important for Cf-2-mediated HR. Hopefully, the authors can develop experimental systems to test Rcr3-Cf-2 interactions to fully address this issue in the future. **RESPONSE:** Yes, the reviewer is right that we cannot distinguish between these scenarios without Cf-2 binding assays. We have expressed this explicitly in the discussion and made clear that this is one of several possible mechanisms.

I hope that the COVID-19 situation there will be better soon. **RESPONSE:** Yes, we hope so

too. Uploading figure files from home takes ages.

Reviewer #2 (Remarks to the Author):

The authors addressed all of my comments. Well done! **RESPONSE:** Thank you!